# Principled Weight Initialisation for Input-Convex Neural Networks

**Pieter-Jan Hoedt & Günter Klambauer**
LIT AI Lab & ELLIS Unit Linz
Institute for Machine Learning
Johannes Kepler University, Linz, Austria
`{hoedt, klambauer}@ml.jku.at`

## Abstract

Input-Convex Neural Networks (ICNNs) are networks that guarantee convexity in their input-output mapping. These networks have been successfully applied for energy-based modelling, optimal transport problems and learning invariances. The convexity of ICNNs is achieved by using non-decreasing convex activation functions and non-negative weights. Because of these peculiarities, previous initialisation strategies, which implicitly assume centred weights, are not effective for ICNNs. By studying signal propagation through layers with non-negative weights, we are able to derive a principled weight initialisation for ICNNs. Concretely, we generalise signal propagation theory by removing the assumption that weights are sampled from a centred distribution. In a set of experiments, we demonstrate that our principled initialisation effectively accelerates learning in ICNNs and leads to better generalisation. Moreover, we find that, in contrast to common belief, ICNNs can be trained without skip-connections when initialised correctly. Finally, we apply ICNNs to a real-world drug discovery task and show that they allow for more effective molecular latent space exploration.

## 1 Introduction

**Input-Convex Networks.** Input-Convex Neural Networks (ICNNs) are networks for which each output neuron is convex with respect to the inputs. The convexity is a result of using non-decreasing convex activation functions and weight matrices with non-negative entries (Amos et al., 2017). ICNNs were originally introduced in the context of energy modelling (Amos et al., 2017). Also in other contexts, ICNNs have proven to be useful. E.g. Sivaprasad et al. (2021) show that ICNNs can be used as regular classification models, Makkuva et al. (2020) rely on the convexity to model optimal transport mappings, and Nesterov et al. (2022) illustrate how ICNNs can be used to learn invariances by simplifying the search for level sets — i.e. inputs for which the output prediction remains unchanged. ICNNs have also been used in model predictive control, where recently an input-convex variant of LSTMs was proposed to reduce convergence time while ensuring closed-loop stability (Wang and Wu, 2023). Despite their successful application for various tasks, convergence can be notably slow (Sivaprasad et al., 2021, Fig. 1 (d)). We hypothesize that this slow training is the result of poor initialisation of the positive weights in ICNNs. This poor initialisation leads to distribution shifts that make learning harder, as illustrated in Figure 1. Therefore, we propose a principled initialisation strategy for layers with non-negative weight matrices (cf. Chang et al., 2020).

**Importance of initialisation strategies.** Initialisation strategies have enabled faster and more stable learning in deep networks (LeCun et al., 1998; Glorot and Bengio, 2010; He et al., 2015). The goal of a good initialisation strategy is to produce similar statistics in every layer. This can be done in the forward pass (LeCun et al., 1998; Mishkin and Matas, 2016; Klambauer et al., 2017; Chang et al.,

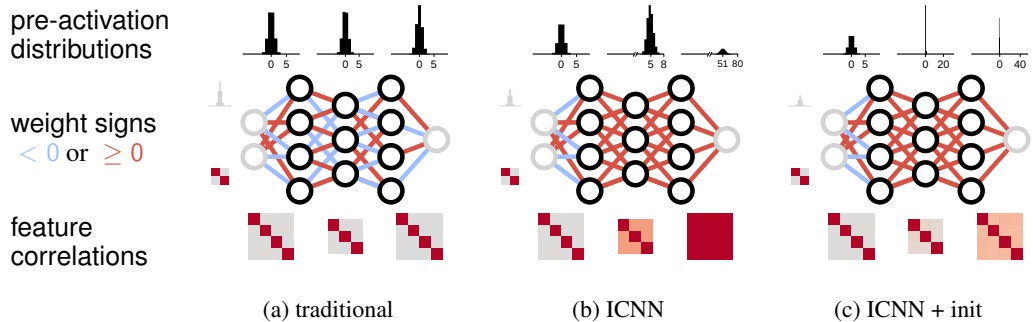

Figure 1: Illustration of the effects due to good signal propagation in hidden layers. Blue and red connections depict negative and positive weights, respectively. The top row shows histograms of pre-activations in each hidden layer. The bottom row displays the feature correlation matrices for these layers. Small visualisations on the left depict the input distribution.

2020) or during back-propagation (Glorot and Bengio, 2010; Hoedt et al., 2018; Defazio and Bottou, 2021). It is also important to account for the effects due to non-linearities in the network (Saxe et al., 2014; He et al., 2015; Klambauer et al., 2017; Hoedt et al., 2018). However, there are no initialisation strategies for non-negative weight matrices, which are a key component of ICNNs (Amos et al., 2017). We derive an initialisation strategy that accounts for the non-negative weights in ICNNs by generalising the signal propagation principles that underlie modern initialisation strategies.

**Signal propagation.** The derivation of initialisation strategies typically builds on the signal propagation framework introduced by Neal (1995). This signal propagation theory has been used and expanded in various ways (Saxe et al., 2014; Poole et al., 2016; Klambauer et al., 2017; Martens et al., 2021). One critical assumption in this traditional signal propagation theory is that weights are sampled from a centred distribution, i.e. with zero mean. In ICNNs, this is not possible because some weight matrices are constrained to be non-negative (Amos et al., 2017). Therefore, we generalise the traditional signal propagation theory to allow for non-centred distributions.

**Skip-connections in ICNNs.** Architectures of ICNNs typically include skip-connections (Amos et al., 2017; Sivaprasad et al., 2021; Makkuva et al., 2020; Nesterov et al., 2022). The skip-connections in ICNNs were introduced to increase their representational power (Amos et al., 2017). Although it is possible to study signal propagation with skip-connections (e.g. Yang and Schoenholz, 2017; Brock et al., 2021; Hoedt et al., 2022), they are typically built on top of existing results for plain networks. Therefore, we limit our theoretical results to ICNNs without skip-connections. However, we find that ICNNs without skip-connections can be successfully trained, indicating that skip-connections might not be necessary for representational power. We show that with our initialisation, we are able to train an ICNN to the same performance as a non-convex baseline without skip-connections. This confirms the hypothesis that good signal propagation can replace skip-connections (Martens et al., 2021; Zhang et al., 2022).

**Contributions.** Our contributions[1] can be summarised as follows:

- We generalise signal propagation theory to include weights without zero mean (Section 2).
- We derive a principled initialisation strategy for ICNNs from our new theory (Section 3).
- We empirically demonstrate the effectiveness of our initialisation strategy (Section 5.1).
- We apply ICNNs in a real-world drug-discovery setting (Sections 5.2 and 5.3).

## 2   Generalising Signal Propagation

In this section, we revisit the traditional signal propagation theory (Neal, 1995), which assumes centred weights. This provides us with a framework to study signal propagation in standard fully-connected networks. We then expand this framework to enable the study of networks where weights do not have zero mean to derive a weight initialisation strategy for ICNNs (Section 3).

---

[1]Code for figures and experiments can be found at https://github.com/ml-jku/convex-init

## 2.1 Background: Traditional Signal Propagation

When studying signal propagation in a network, the effect of a neural network layer on characteristics of the signal, such as its mean or variance, are investigated. The analysis of the vanishing gradient problem (Hochreiter, 1991) can also be considered as signal propagation theory where the norm of the delta-errors is tracked throughout layers in backpropagation. Typically, the network is assumed to be a repetitive composition of the same layers (e.g. fully-connected or convolutional). This allows to reduce the analysis of an entire network to a single layer and enables fixed-point analyses (e.g. Klambauer et al., 2017; Schoenholz et al., 2017). In our work, we focus on fully-connected networks, $f : \mathbb{R}^N \to \mathbb{R}^M$, with some activation function $\phi : \mathbb{R} \to \mathbb{R}$ that is applied element-wise to vectors. We study the propagation of pre-activations[2] (cf. Saxe et al., 2014; He et al., 2015; Poole et al., 2016) throughout the network:

$$s = W\phi(s^-) + b. \tag{1}$$

Here, $W \in \mathbb{R}^{M \times N}$ and $b \in \mathbb{R}^M$ are the weight matrix and bias vector, respectively. The notation $s^-$ is used to indicate the pre-activations from the preceding layer, such that $s^- \in \mathbb{R}^N$.

The first two moments of the signal can be expressed in terms of the randomness arising from the parameters. At initialisation time, the weight parameters are considered identically and independently distributed (i.i.d.) random variables $w_{ij} \sim \mathcal{D}_w$. The weight distribution is often assumed to be uniform or Gaussian, but the exact shape does not matter in wide networks (Golikov and Yang, 2022, Principle 2). The bias parameters, on the other hand, are commonly ignored in the analysis, which is justified when $\forall i : b_i = 0$. The pre-activations are also random variables due to the randomness of the parameters. If we assume the weights to be centred, such that $\mathbb{E}[w_{ij}] = 0$, and $\mathrm{Var}[w_{ij}] = \sigma_w^2$, the first two moments of the pre-activations are given by

$$\mathbb{E}[s_i] = \mathbb{E}\left[\sum_k w_{ik}\phi(s_k^-) + b_i\right] = 0 \tag{2}$$

$$\mathbb{E}[s_i^2] = \mathbb{E}\left[\left(\sum_k w_{ik}\phi(s_k^-) + b_i\right)^2\right] = N\sigma_w^2 \mathbb{E}[\phi(s_1^-)^2]. \tag{3}$$

Note that the variance is independent of the index, $i$, in the pre-activation vector and thus $\forall k : \mathbb{E}[\phi(s_k^-)^2] = \mathbb{E}[\phi(s_1^-)^2]$. Moreover, it can be shown (see Appendix A) that $\forall i \neq j : \mathbb{E}[s_i s_j] = 0$, i.e. features within a pre-activation vector are uncorrelated in expectation.

Signal propagation theory has been used to derive initialisation and normalisation methods (LeCun et al., 1998; Glorot and Bengio, 2010; He et al., 2015; Klambauer et al., 2017). Initialisation methods often aim at having pre-activations with the same mean and variance in every layer of the network. Because the mean is expected to be zero in every layer, the focus lies on keeping the variance of the pre-activations constant throughout the network, i.e. $\forall i, j : \mathbb{E}[s_i^2] = \mathbb{E}[s_j^{-2}] = \sigma_*^2$. Plugging this into Eq. (3), we obtain the fixed-point equation

$$\sigma_*^2 = N\sigma_w^2 \mathrm{varprop}_\phi(\sigma_*^2). \tag{4}$$

Here, $\mathrm{varprop}_\phi$ is a function that models the propagation of variance through the activation function, $\phi$, assuming zero mean inputs. E.g. $\mathrm{varprop}_{\mathrm{ReLU}}(\sigma^2) = \frac{1}{2}\sigma^2$ (He et al., 2015) or $\mathrm{varprop}_{\tanh}(\sigma^2) \approx \sigma^2$ (LeCun et al., 1998; Glorot and Bengio, 2010). We refer to Appendix A.3 for a more detailed discussion on propagation through activation functions.

This shows how modern initialisation methods are a solution to a fixed-point equation of the variance propagation. Our goal is to apply the same principles to find an initialisation strategy for the non-negative weights in ICNNs. However, Eq. (4) heavily relies on the assumption that the weights are centred, i.e. $\mu_w = 0$. This assumption is impossible to satisfy for the non-negative weights in ICNNs, unless all weights are set to zero. If $\mu_w \neq 0$, the mean of the pre-activations is no longer zero by default (as in Eq. 2). Furthermore, also the propagation of variance and covariance is affected. Therefore, we extend the signal propagation theory to accurately describe the effects due to non-centred weights.

---

[2]a common alternative is to consider the (post-)activations (see Hoedt et al., 2018)

## 2.2 Generalised Signal Propagation

We generalise the traditional signal propagation theory by lifting a few assumptions from the traditional approach. Most notably, the weights are no longer drawn from a zero-mean distribution, such that $\mathbb{E}[w_{ij}] = \mu_w \neq 0$. Additionally, we include the effects due to bias parameters, which will give us extra options to control the signal propagation. Similar to the weights, we assume bias parameters to be i.i.d. samples from some distribution with mean $\mathbb{E}[b_i] = \mu_b$ and variance $\mathrm{Var}[b_i] = \sigma_b^2$. By re-evaluating the expectations in Eq. (2) and Eq.(3), we arrive at the following results:

$$\mathbb{E}[s_i] = N\mu_w \mathbb{E}[\phi(s_1^-)] + \mu_b \tag{5}$$

$$\mathbb{E}[s_i s_j] = \delta_{ij}\left(N\sigma_w^2 \mathbb{E}\left[\phi(s_1^-)^2\right] + \sigma_b^2\right) + \mu_w^2 \sum_{k,k'} \mathrm{Cov}[\phi(s_k^-), \phi(s_{k'}^-)] + \mathbb{E}[s_i]\,\mathbb{E}[s_j], \tag{6}$$

where $\delta_{ij}$ is the Kronecker delta. We refer to Appendix A.2 for a full derivation. Note that mean and variance are still independent of the index, $i$. Therefore, we can continue to assume $\forall k$ : $\mathbb{E}\left[\phi(s_k^-)^n\right] = \mathbb{E}\left[\phi(s_1^-)^n\right]$.

There are a few crucial differences between the traditional approach and our generalisation. First, the expected value of the pre-activations is no longer zero by default. This also means that the second moment, $\mathbb{E}\left[s_i^2\right]$, does not directly model the variance of the pre-activations in our setting. Secondly, the propagation of the second moment has an additional term that incorporates effects due to the covariance structure of pre-activations in the previous layer. Finally, the off-diagonal elements of the feature covariance matrix, $\mathrm{Cov}[s_i, s_j] = \mathbb{E}_{i \neq j}[s_i s_j] - \mathbb{E}[s_i]\,\mathbb{E}[s_j]$, are not necessarily zero and also depend on the variance. This interaction between on- and off-diagonal elements in the feature covariance matrix makes it harder to ensure stable propagation.

Similar to Eq. (3), Eq. (6) depends on the variance propagation through the activation function. In addition, our generalised propagation theory involves the covariance propagation through the activation function. The covariance propagation through the ReLU function has been studied in prior work (Cho and Saul, 2009; Daniely et al., 2016) and can be specified as follows:

$$\mathbb{E}[\mathrm{ReLU}(s_1)\,\mathrm{ReLU}(s_2)] = \frac{\sigma^2}{2\pi}\left(\sqrt{1 - \rho^2} + \rho \arccos(-\rho)\right), \tag{7}$$

with $(s_1, s_2) \sim \mathcal{N}\left((0,0), \begin{pmatrix} 1 & \rho \\ \rho & 1 \end{pmatrix}\sigma^2\right)$ and correlation $\rho$. Note that this expression implicitly describes the propagation of the squared mean ($\rho = 0$) and the second raw moment ($\rho = 1$). We will assume $\phi = \mathrm{ReLU}$ for the remainder of our analysis. A derivation for the propagation through the leaky ReLU activation function (Maas et al., 2013) can be found in Appendix A.3.

With our generalised propagation theory, the effects due to $\mu_w \neq 0$ become apparent. As expected, the pre-activations no longer have zero mean by default. Furthermore, the covariance plays a significant role in the signal propagation. This shows that we also have to stabilise mean and covariance on top of the variance propagation to derive our principled initialisation.

## 3 Principled Initialisation of ICNNs

With our generalised signal propagation theory, we will now derive a principled initialisation strategy for ICNNs. First, we set up fixed point equations for mean, variance and correlation using our generalised framework. By solving these fixed point equations for $\mu_w, \sigma_w^2, \mu_b$ and $\sigma_b^2$ we obtain our principled initialisation.

### 3.1 Background: Input-Convex Neural Networks

Neural networks for which all input-output mappings are convex, are called Input-Convex Neural Networks (ICNNs) (Amos et al., 2017). In general, neural networks are functions that are constructed by composing one or more simpler layers or components. A function composition $f \circ g$ is convex if both $f$ and $g$ are convex, where $f$ additionally has to be non-decreasing in every output. Therefore, ICNNs are created by enforcing that every layer is *convex* and *non-decreasing* (Amos et al., 2017). A direct consequence is that only convex, non-decreasing activation functions can be

used in ICNNs — e.g. softplus (Dugas et al., 2001), ReLU (Nair and Hinton, 2010), LReLU (Maas et al., 2013), ELU (Clevert et al., 2016).

Fully-connected and convolutional layers are affine transformations, which are trivially convex. To make these layers non-decreasing for building ICNNs, their weights have to be non-negative (Amos et al., 2017). This can be done by replacing negative values with zero after every update. The non-negativity constraint does not apply to bias parameters. Another notable exception are direct connections from the input layer (Amos et al., 2017). This means that the first layer and any skip-connections from the input can have unconstrained weights. Amos et al. (2017) argue that these skip-connections are necessary for representational power. We limit our analysis to networks without skip-connections. Our experiments (see Section 5) show that ICNNs with our proposed initialisation do not require skip-connections for efficient training.

## 3.2 (Co-)Variance Fixed Points

To set up the fixed-point equation for the second moments (Eq. 6), we need to be able to propagate through the ReLU non-linearity. However, the analytical results from Eq. (7) only hold if the pre-activations have zero mean. Therefore, we look for a configuration that makes the mean (Eq. 5) of the pre-activations zero. Because we cannot enforce $\mu_w = 0$, we make use of the bias parameters to obtain centred pre-activations:

$$\mu_b = -N\mu_w \, \mathbb{E}[\phi(s_1^-)]. \tag{8}$$

This also allows us to refer to the second moments as the (co-)variance. Note that we approximate the pre-activations with Gaussians to use Eq. (7) at this point and for the remainder of the analysis. Figure 1c suggests that this Gaussian assumption generally does not hold in ICNNs (see also Appendix A.3). Nevertheless, our experiments (see Section 5) suggest that the approximation is sufficiently good to derive an improved initialisation strategy for ICNNs. Furthermore, Figure 6b indicates that the use of random initial bias parameters (cf. Appendix C.3) causes pre-activations to be (slightly) more Gaussian.

The propagation of the second moment described by Eq. (6) consists of two parts. The first part describes the off-diagonal entries, for which the dynamics are given by

$$\text{Cov}_{i \neq j}[s_i, s_j] = \mu_w^2 \sum_{k,k'} \text{Cov}[\phi(s_k^-), \phi(s_{k'}^-)] = N\mu_w^2 \Big( \text{Var}[\phi(s_1^-)] + (N-1)\,\text{Cov}[\phi(s_1^-), \phi(s_2^-)] \Big).$$

Here, we rewrite the sum assuming that the pre-activations are identically distributed. This is possible because the covariance does not depend on the indices $i$ or $j$. For further details, we refer to Appendix A.2 The second part models the on-diagonal entries, i.e. the variance, which can be simplified in a similar way

$$\text{Var}[s_i] = N\sigma_w^2 \, \mathbb{E}[\phi(s_1^-)] + \sigma_b^2 + \text{Cov}[s_1, s_2].$$

By plugging the ReLU moments from Eq. (7) into these results in Appendix B.2, we obtain the desired fixed-point equations in terms of variance and correlation:

$$\rho_* = \mu_w^2 \frac{1}{2\pi} N \Big( \pi - N + (N-1)\big(\sqrt{1 - \rho_*^2} + \rho_* \arccos(-\rho_*)\big) \Big) \tag{9}$$

$$\sigma_*^2 = N\sigma_w^2 \frac{1}{2} \sigma_*^2 + \sigma_b^2 + \rho_* \sigma_*^2. \tag{10}$$

with $\rho_* = \frac{1}{\sigma_*^2} \text{Cov}[s_1^-, s_2^-] = \frac{1}{\sigma_*^2} \text{Cov}_{i \neq j}[s_i, s_j]$ and $\sigma_*^2 = \text{Var}[s_1^-] = \text{Var}[s_i]$.

## 3.3 Weight Distribution for ICNNs

To obtain the distribution parameters for the initial weights, we solve the fixed point equations in Eq. (9) and Eq.(10) for $\sigma_b^2$, $\sigma_w^2$ and $\mu_w^2$. Because this system is over-parameterised, we choose to set $\sigma_b^2 = 0$. An analysis for $\sigma_b^2 > 0$ can be found in Appendix C.3. This leads to the following initialisation parameters for ICNNs:

$$\sigma_w^2 = \frac{2}{N}(1 - \rho_*) \tag{11}$$

$$\mu_w^2 = \frac{2\pi}{N} \rho_* \Big( \pi - N + (N-1)\big(\sqrt{1 - \rho_*^2} + \rho_* \arccos(-\rho_*)\big) \Big)^{-1}. \tag{12}$$

Note that these solutions only depend on the correlation, $\rho_*$, and not on the variance, $\sigma_*^2$. For a stability analysis of these fixed points, we refer to Appendix B.4.

Although our initialisation (Eq. 12 and 11) is entirely defined by the correlation between features, $\rho = \frac{\text{Cov}[s_1, s_2]}{\text{Var}[s_1]}$, not all solutions are admissible. For example, uncorrelated features ($\rho = 0$) are only possible if $\mu_w = 0$. This means that features in ICNNs must have non-zero correlation. On the other hand, perfect correlation ($\rho = 1$) would mean that all features point in the same direction, which is not desirable. Therefore, we choose $\rho_* = \frac{1}{2}$ as a compromise to obtain the following initialisation parameters for our experiments in Section 5:

$$\mu_w = \sqrt{\frac{6\pi}{N\left(6(\pi-1)+(N-1)(3\sqrt{3}+2\pi-6)\right)}} \qquad \sigma_w^2 = \frac{1}{N}$$

$$\mu_b = \sqrt{\frac{3N}{6(\pi-1)+(N-1)(3\sqrt{3}+2\pi-6)}} \qquad \sigma_b^2 = 0.$$

We refer to appendix C.4 for a discussion on different choices for $\rho_*$.

For the first layer, which can also have negative weights, we use LeCun et al. (1998) initialisation. For the non-negative weights in an ICNNs, we need to sample from a distribution with non-negative support. The lower bound of a uniform distribution with our suggested mean and variance is negative for any $N > 1$ and thus not useful. Also, sampling from a Gaussian distribution is not practical because its support includes negative values. Eventually, we propose to sample from a log-normal distribution with parameters

$$\tilde{\mu_w} = \ln(\mu_w^2) - \frac{1}{2}\ln(\sigma_w^2 + \mu_w^2) \qquad \tilde{\sigma_w}^2 = \ln(\sigma_w^2 + \mu_w^2) - \ln(\mu_w^2).$$

This ensures that sampled weights are non-negative and have the desired mean and variance. Note that other positive distributions with sufficient degrees of freedom should also work.

One advantage of the log-normal distribution is that sampling is simple and efficient. It suffices to sample $\tilde{w}_{ij} \sim \mathcal{N}(\tilde{\mu_w}, \tilde{\sigma_w}^2)$ to compute $w_{ij} = \exp(\tilde{w}_{ij})$. In the original ICNN (Amos et al., 2017), the exponential function is only used to make the initial weights positive. However, if weights are re-parameterised using the exponential function, initial weights can be directly sampled from a Gaussian distribution. This setting is studied in Appendix C.5.

## 4 Related Work

*Input-convex neural networks.* Input-Convex Neural Network (ICNN) were originally designed in the context of energy models (Amos et al., 2017). We focus on the fully convex ICNNs variant and use gradient descent for optimisation instead of the proposed bundle-entropy method. In their implementation, Amos et al. (2017) use projection methods to keep weights positive, i.e. negative values are set to zero after every update. However, also other projection methods can be used to keep the weights positive (e.g. Sivaprasad et al., 2021). Instead of projecting the weights onto the non-negative reals after every update, it is also possible to use a reparameterisation of the weights. E.g. Nesterov et al. (2022) square the weights in the forward pass. Note that a reparameterisation can have an effect on the learning dynamics because it is an inherent part of the forward pass. Another alternative is to use regularisation to impose a *soft* non-negativity constraint on the weights (Makkuva et al., 2020). Similar to (Sivaprasad et al., 2021), we directly train an ICNN, but we do not observe the same generalisation benefits. On the other hand, Sankaranarayanan and Rengaswamy (2022) point out that restricting the network to be convex decreases their capacity. They propose to use the difference of two ICNNs to allow modelling non-convex functions (c.f. Yuille and Rangarajan, 2001). However, we did not find the convexity restriction to cause any problems in our experiments. Nesterov et al. (2022) use ICNNs to simplify the computation of level sets to learn invariances. We adopt this experiment setting to highlight the potential of ICNNs. *Optimization.* While training the parameters of an ICNN is unrelated to convex optimisation, Bengio et al. (2005) showed how training a neural network together with the number of hidden neurons can be seen as a convex optimisation problem. This work was continued by Bach (2017), who established a connection between this convex optimisation and automatic feature selection. *Signal propagation theory.* The study of signal propagation is concerned with tracking statistics of the data throughout multiple

layers of the network. In his thesis, Neal (1995) computed how the first two moments propagate through a two-layer network to study networks as Gaussian processes. This is similar to how modern initialisation strategies have been derived (e.g. LeCun et al., 1998; He et al., 2015). A similar approach has been taken to incorporate the propagation of gradients for initialisation (Glorot and Bengio, 2010). The idea to explicitly account for the effects of activation functions can be attributed to (Saxe et al., 2014). All of these analyses assume that weights are initialised from a zero-mean distribution, which is not possible in ICNNs. Mishkin and Matas (2016) empirically compute the variance for the weights. This approach is more robust to deviations, but nevertheless requires adaptations to account for weights with non-zero mean. There is also a significant body of work that studies signal propagation in terms of mean field theory (Poole et al., 2016). In these works, the correlation between samples is included in the analysis (Poole et al., 2016). In our work, we find that the correlation between features plays an important role. Schoenholz et al. (2017) applied the mean field analysis to the backward dynamics and introduced the concept of *depth scales* to quantify how deep signals can propagate. The mean field theory was also applied to convolutional networks to derive the delta-orthogonal initialisation (Xiao et al., 2018). We refer to (Martens et al., 2021) for a general overview of signal propagation.

# 5 Experiments

We evaluate our principled initialisation strategy by training ICNNs on three sets of experiments. In our first experiments, we investigate the effect of initialisation on learning dynamics and generalisation in ICNNs on multiple permuted image datasets. We also include non-convex networks in these experiments to illustrate that ICNNs with our principled initialisation can be trained as well as regular networks. However, we would like to stress that non-convex networks are expected to outperform ICNNs because they are not constrained to convex decision boundaries. Moreover, we assess whether ICNNs using our initialisation are able to match the performance of regular fully-connected networks in toxicity prediction without the need for skip-connections. Finally, we explore ICNNs as a tool for more controlled latent space exploration of a molecule auto-encoder system. This illustrates a setting where regular networks can no longer be used. Details on the computing budget are provided in the Appendix.

## 5.1 Computer vision benchmarking datasets

As an illustrative first set of experiments, we use computer vision benchmarks to assess the effects of our principled initialisation strategy. Concretely, we trained fully-connected ICNNs on MNIST (Bottou et al., 1994), CIFAR10 and CIFAR100 (Krizhevsky, 2009), to which we refer as permuted image benchmarks (cf. Goodfellow et al., 2014).

In our comparison, we consider four different types of methods: (1) a classical non-convex network without skip-connections and default initialisation (cf. He et al., 2015). (2) an ICNN, "ICNN", without skip connections and default initialisation (3) an ICNN, "ICNN + skip", with skip connections and default initialisation (4) an ICNN, "ICNN + init", without skip connections and our proposed initialisation. All networks use ReLU activation functions. The non-negativity constraint in ICNNs is implemented by setting negative weights to zero after every update (cf. Amos et al., 2017). Note that skip-connections in ICNNs introduce additional parameters to allow for additional representational power (cf. Amos et al., 2017).

**Training dynamics.** *Settings.* In the first experiment, we analyze the training dynamics during the first ten epochs of training. Similar to Chang et al. (2020), we fixed the number of neurons in each layer to the input size, and the number of hidden layers to five. We refer to Appendix C.2 for results with different depths. The learning rate for the Adam optimiser was obtained by manually tuning on the non-convex baseline. *Results.* We found that ICNNs with our principled initialisation exhibit similar learning behaviour as the non-convex baseline, while ICNNs without our initialisation strategy can not decrease the loss to the level of the baseline on CIFAR10 and CIFAR100. Figure 2 shows the learning curves for our different methods on three permuted image benchmarks.

**Generalisation.** *Settings.* In our next experiment, the effects of our principled initialisation on the generalisation capabilities of ICNNs is investigated. To this end, we train our three ICNN variants and the non-convex baseline on the permuted image benchmarks again, but focus on the test performance this time. To this end we perform a grid-search to find the hyper-parameters that attain the

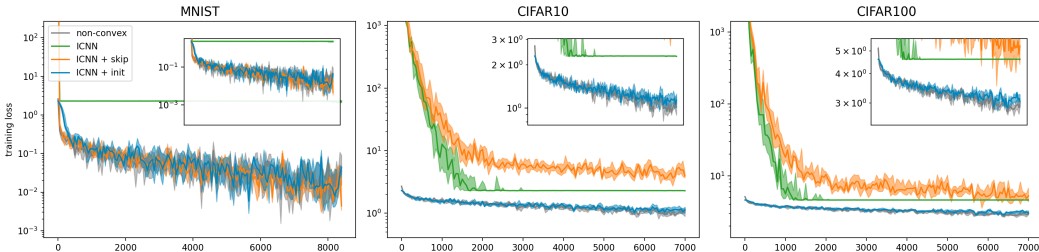

Figure 2: Training loss curves of ICNN variants with the same architecture on the MNIST, CIFAR10 and CIFAR100 datasets. "ICNN" input-convex network with default initialisation. "ICNN + skip": same settings, but with skip connections. "ICNN + init" our principled initialisation for ICNNs w/o skip connections. "non-convex": a regular non-convex network. Each curve represents the median performance over ten runs and shaded regions indicate the inter-quartile range. The inset figures provide a view of the loss curves zoomed in. Note that ICNN losses do decrease before the plateau.

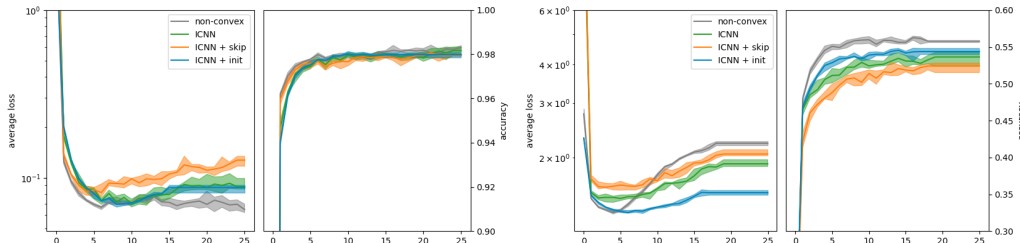

Figure 3: Test set metrics of compared methods on the (a) MNIST and (b) CIFAR10 datasets. Each curve displays the median performance over ten runs. Shaded regions represent the inter-quartile range over the ten runs. On MNIST, all methods can be successfully trained and exhibit similar performance. On CIFAR 10, ICNNs with our proposed initialisation outperform other ICNNs variants.

best accuracy after 25 epochs of training on a random validation split, for each of the four compared methods. The grid consists of multiple architectures with at least one hidden layer, learning rate for Adam, $L_2$-regularisation and whitening pre-processing transforms (details in Appendix C.2). *Results.* On the MNIST dataset, all variants reach similar accuracy values, which we attribute to the simplicity of the prediction problem. However, on CIFAR10 the initialisation strategy leads to better generalisation already in early epochs and — compared to the respective ICNNs variants without our initialisation — improves generalisation overall (see Figure 3). The ICNN with our initialisation almost matches the accuracy values of non-convex nets.

## 5.2 Toxicity Prediction

To test ICNNs in a real-world setting, in a different application domain, we consider the binary multi-task problem of toxicity prediction in drug discovery. More specifically, we train ICNNs on the Tox21 challenge data (Huang et al., 2016; Mayr et al., 2016; Klambauer et al., 2017). The input data consists of so-called SMILES representations of small molecules. The target variables are binary labels for twelve different measurements or assays indicating whether a molecule induces a particular toxic effect on human cells. During pre-processing, SMILES strings were converted to Continuous and Data-Driven Descriptors (CDDDs) (Winter et al., 2019), which are 512-dimensional numerical representations of the molecules. The choice of these particular descriptors is motivated by CDDDs properties of being decodeable into valid molecules (see Section 5.3). Hyper-parameters were selected by a manual search on the non-convex baseline. We ended up using a fully-connected network with two hidden layers of 128 neurons and ReLU activations. The network was regularised with fifty percent dropout after every hidden layer, as well as seventy percent dropout of the inputs. The results in Table 1 show that our initialisation significantly outperforms ICNNs with standard initialization ($p$-value 2.6e-13, binomial test) and ICNNs with skip connections ($p$-value 5.5e-14). Furthermore, results are close to the performance of traditional, non-convex networks.

Table 1: Area under the ROC curve on the test set for each of the twelve tasks in the Tox21 data. Each value represents the median performance over 10 runs and error bars is the maximum distance to the boundary of the interval defined by the (0.05, 0.95) quantiles. Of all variants of ICNNs, the variant with our proposed initialisation, "ICNN+init (ours)", performs best matching almost the predictive quality of traditional non-convex neural networks.

| | NR.AhR | NR.AR | NR.AR.LBD | NR.Aromatase | NR.ER | NR.ER.LBD |
|---|---|---|---|---|---|---|
| ICNN+init (ours) | **91.29** ± 0.58% | **81.84** ± 2.43% | 81.36 ± 3.58% | 82.40 ± 1.09% | **78.37** ± 0.80% | 77.85 ± 0.90% |
| ICNN | 90.56 ± 0.63% | 81.28 ± 4.82% | 78.14 ± 3.15% | 80.46 ± 1.74% | 77.30 ± 0.78% | 78.15 ± 1.02% |
| ICNN+skip | 89.83 ± 0.21% | 68.12 ± 1.74% | 74.17 ± 2.20% | 78.95 ± 0.45% | 76.95 ± 0.48% | **81.92** ± 1.16% |
| non-convex | 91.01 ± 0.65% | 79.21 ± 1.73% | **86.19** ± 2.50% | **83.63** ± 0.34% | 77.88 ± 1.13% | 74.32 ± 1.59% |

| | SR.ATAD5 | SR.HSE | SR.MMP | SR.p53 | NR.PPARγ | SR.ARE | AVG |
|---|---|---|---|---|---|---|---|
| ICNN+init (ours) | 77.01 ± 1.67% | **80.05** ± 1.36% | 93.69 ± 0.27% | 81.10 ± 0.46% | 77.46 ± 2.50% | 76.80 ± 0.33% | 80.57 ± 11.84% |
| ICNN | 74.25 ± 3.26% | 78.64 ± 1.42% | 93.19 ± 0.59% | 81.32 ± 0.43% | 75.00 ± 1.73% | 76.11 ± 0.82% | 78.33 ± 13.42% |
| ICNN+skip | 75.93 ± 1.26% | 78.23 ± 0.50% | 75.40 ± 0.88% | 78.25 ± 0.93% | **92.09** ± 0.41% | **79.32** ± 0.97% | 78.29 ± 12.56% |
| non-convex | **78.93** ± 1.68% | 80.12 ± 2.57% | **93.99** ± 0.39% | **82.17** ± 0.96% | **82.70** ± 1.21% | 78.30 ± 0.90% | **81.31** ± 11.03% |

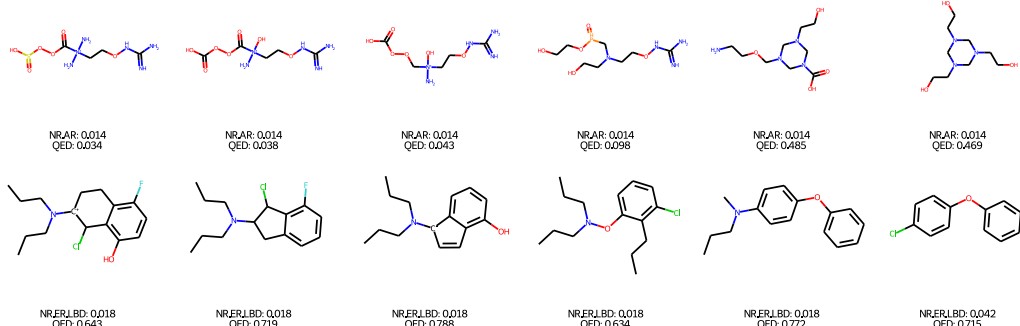

Figure 4: Example of two level set trajectories of a Tox21 model. The leftmost molecule represents the starting molecule with low toxicity (top: "NR.AR", bottom: "NR.ER.LBD"). The rightmost molecule represents the target molecule. All intermediary molecules are samples on the level set between the starting molecule and the target molecule and exhibit different drug-likeness ("QED"). In this way, one predicted molecular property can be kept fixed while another property is optimized.

### 5.3 Latent space exploration of molecular spaces

In this experiment, we exploit the intrinsic property of ICNNs that level sets, i.e. the set of all inputs that map to the same output, can be parameterised (Nesterov et al., 2022). These level sets provide an opportunity in drug discovery to keep one molecular property, such as low toxicity, fixed while optimizing another property, such as drug-likeness. To demonstrate this application, we randomly chose reference molecules with low toxicity and followed the according level set in the direction of another randomly chosen target molecules. Since the input space of the ICNN that predicts toxicity (see above) possesses a decoder model from (Winter et al., 2019), the numeric representations of the molecules on the level set can be decoded into molecular structures. The obtained trajectories in Figure 4 demonstrate that the predicted toxicity of the molecules remain constant, while different QED scores, which measure drug-likeness, are obtained by traversing the level set from the reference molecule to the target molecule.

## 6 Conclusion and Discussion

We demonstrated how signal propagation theory can be generalised to include weights without zero mean. By controlling the propagation of correlation through the network, we derived an initialisation strategy that promotes a stable propagation of signals. We empirically verified that this steady propagation leads to improved learning. Finally, we showed that ICNNs can be trained to a comparable level as regular networks on the Tox21 data using CDDDs.

Our initialisation approximates the distribution of pre-activations with Gaussians, which falls short of agreement with the observed data, but has been sufficient to improve the initialisation of ICNNs. A possible way forward is to use the central limit theorem for weakly dependent variables. In Ap-

pendix C.3, we also observed that random bias initialisation leads to pre-activations with a more Gaussian distribution. Appendix A.5 also includes an analysis for convolutional layers, but proficient empirical results with input-convex convolutional networks are still to be made. We conjecture that this is due to the more complicated covariance structure in convolutional layers. We also included a signal propagation analysis for back-propagation in Appendix A.4. However, incorporating insights from this analysis into an initialisation method is also left for future work. We envision that our initialisation strategy will make it easier to incorporate ICNNs or other networks with non-negative weights.

## Acknowledgments and Disclosure of Funding

We thank Sepp Hochreiter for pointing us in the direction of input-convex networks and their interesting properties. Special thanks go to Niklas Schmidinger for his help with experiments on convolutional networks.

The ELLIS Unit Linz, the LIT AI Lab, the Institute for Machine Learning, are supported by the Federal State Upper Austria. We thank the projects AI-MOTION (LIT-2018-6-YOU-212), DeepFlood (LIT-2019-8-YOU-213), Medical Cognitive Computing Center (MC3), INCONTROL-RL (FFG-881064), PRIMAL (FFG-873979), S3AI (FFG-872172), DL for GranularFlow (FFG-871302), EPILEPSIA (FFG-892171), AIRI FG 9-N (FWF-36284, FWF-36235), AI4GreenHeatingGrids(FFG- 899943), INTEGRATE (FFG-892418), ELISE (H2020-ICT-2019-3 ID: 951847), Stars4Waters (HORIZON-CL6-2021-CLIMATE-01-01). We thank Audi.JKU Deep Learning Center, TGW LOGISTICS GROUP GMBH, Silicon Austria Labs (SAL), FILL Gesellschaft mbH, Anyline GmbH, Google, ZF Friedrichshafen AG, Robert Bosch GmbH, UCB Biopharma SRL, Merck Healthcare KGaA, Verbund AG, GLS (Univ. Waterloo) Software Competence Center Hagenberg GmbH, TÜV Austria, Frauscher Sensonic, TRUMPF and the NVIDIA Corporation.

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
