| ICNN | $90.56 \pm 0.63\%$ | $81.28 \pm 4.82\%$ | $78.14 \pm 3.15\%$ | $80.46 \pm 1.74\%$ | $77.30 \pm 0.78\%$ | $78.15 \pm 1.02\%$ |
| ICNN+skip | $89.83 \pm 0.21\%$ | $68.12 \pm 1.74\%$ | $74.17 \pm 2.20\%$ | $78.95 \pm 0.45\%$ | $76.95 \pm 0.48\%$ | $\mathbf{81.92} \pm 1.16\%$ |
| non-convex | $91.01 \pm 0.65\%$ | $79.21 \pm 1.73\%$ | $\mathbf{86.19} \pm 2.50\%$ | $\mathbf{83.63} \pm 0.34\%$ | $77.88 \pm 1.13\%$ | $74.32 \pm 1.59\%$ |

| | SR.ATAD5 | SR.HSE | SR.MMP | SR.p53 | NR.PPAR$\gamma$ | SR.ARE | AVG |
|---|---|---|---|---|---|---|---|
| ICNN+init (ours) | $77.01 \pm 1.67\%$ | $\mathbf{80.05} \pm 1.36\%$ | $93.69 \pm 0.27\%$ | $81.10 \pm 0.46\%$ | $77.46 \pm 2.50\%$ | $76.80 \pm 0.33\%$ | $80.57 \pm 11.84\%$ |
| ICNN | $74.25 \pm 3.26\%$ | $78.64 \pm 1.42\%$ | $93.19 \pm 0.59\%$ | $81.32 \pm 0.43\%$ | $75.00 \pm 1.73\%$ | $76.11 \pm 0.82\%$ | $78.33 \pm 13.42\%$ |
| ICNN+skip | $75.93 \pm 1.26\%$ | $78.23 \pm 0.50\%$ | $75.40 \pm 0.88\%$ | $78.25 \pm 0.93\%$ | $\mathbf{92.09} \pm 0.41\%$ | $\mathbf{79.32} \pm 0.97\%$ | $78.29 \pm 12.56\%$ |
| non-convex | $\mathbf{78.93} \pm 1.68\%$ | $\mathbf{80.12} \pm 2.57\%$ | $\mathbf{93.99} \pm 0.39\%$ | $\mathbf{82.17} \pm 0.96\%$ | $82.70 \pm 1.21\%$ | $78.30 \pm 0.90\%$ | $\mathbf{81.31} \pm 11.03\%$ |

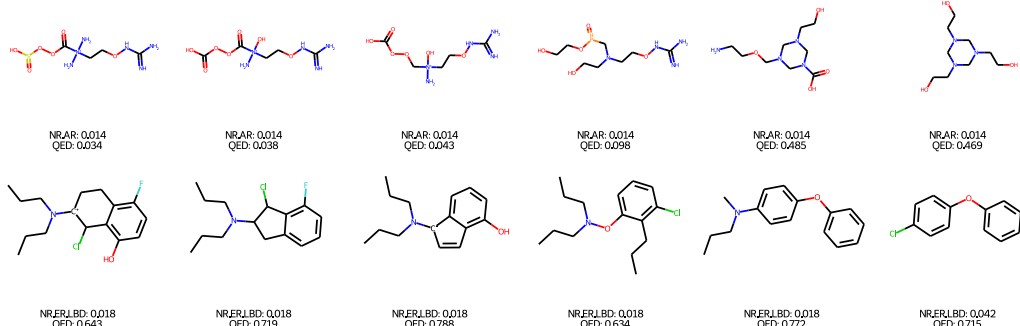

Figure 4: Example of two level set trajectories of a Tox21 model. The leftmost molecule represents the starting molecule with low toxicity (top: "NR.AR", bottom: "NR.ER.LBD"). The rightmost molecule represents the target molecule. All intermediary molecules are samples on the level set between the starting molecule and the target molecule and exhibit different drug-likeness ("QED"). In this way, one predicted molecular property can be kept fixed while another property is optimized.

## 5.3 Latent space exploration of molecular spaces

In this experiment, we exploit the intrinsic property of ICNNs that level sets, i.e. the set of all inputs that map to the same output, can be parameterised (Nesterov et al., 2022). These level sets provide an opportunity in drug discovery to keep one molecular property, such as low toxicity, fixed while optimizing another property, such as drug-likeness. To demonstrate this application, we randomly chose reference molecules with low toxicity and followed the according level set in the direction of another randomly chosen target molecules. Since the input space of the ICNN that predicts toxicity (see above) possesses a decoder model from (Winter et al., 2019), the numeric representations of the molecules on the level set can be decoded into molecular structures. The obtained trajectories in Figure 4 demonstrate that the predicted toxicity of the molecules remain constant, while different QED scores, which measure drug-likeness, are obtained by traversing the level set from the reference molecule to the target molecule.

## 6 Conclusion and Discussion

We demonstrated how signal propagation theory can be generalised to include weights without zero mean. By controlling the propagation of correlation through the network, we derived an initialisation strategy that promotes a stable propagation of signals. We empirically verified that this steady propagation leads to improved learning. Finally, we showed that ICNNs can be trained to a comparable level as regular networks on the Tox21 data using CDDDs.

Our initialisation approximates the distribution of pre-activations with Gaussians, which falls short of agreement with the observed data, but has been sufficient to improve the initialisation of ICNNs. A possible way forward is to use the central limit theorem for weakly dependent variables. In Ap-

pendix C.3, we also observed that random bias initialisation leads to pre-activations with a more Gaussian distribution. Appendix A.5 also includes an analysis for convolutional layers, but proficient empirical results with input-convex convolutional networks are still to be made. We conjecture that this is due to the more complicated covariance structure in convolutional layers. We also included a signal propagation analysis for back-propagation in Appendix A.4. However, incorporating insights from this analysis into an initialisation method is also left for future work. We envision that our initialisation strategy will make it easier to incorporate ICNNs or other networks with non-negative weights.

## Acknowledgments and Disclosure of Funding

We thank Sepp Hochreiter for pointing us in the direction of input-convex networks and their interesting properties. Special thanks go to Niklas Schmidinger for his help with experiments on convolutional networks.

The ELLIS Unit Linz, the LIT AI Lab, the Institute for Machine Learning, are supported by the Federal State Upper Austria. We thank the projects AI-MOTION (LIT-2018-6-YOU-212), DeepFlood (LIT-2019-8-YOU-213), Medical Cognitive Computing Center (MC3), INCONTROL-RL (FFG-881064), PRIMAL (FFG-873979), S3AI (FFG-872172), DL for GranularFlow (FFG-871302), EPILEPSIA (FFG-892171), AIRI FG 9-N (FWF-36284, FWF-36235), AI4GreenHeatingGrids(FFG- 899943), INTEGRATE (FFG-892418), ELISE (H2020-ICT-2019-3 ID: 951847), Stars4Waters (HORIZON-CL6-2021-CLIMATE-01-01). We thank Audi.JKU Deep Learning Center, TGW LOGISTICS GROUP GMBH, Silicon Austria Labs (SAL), FILL Gesellschaft mbH, Anyline GmbH, Google, ZF Friedrichshafen AG, Robert Bosch GmbH, UCB Biopharma SRL, Merck Healthcare KGaA, Verbund AG, GLS (Univ. Waterloo) Software Competence Center Hagenberg GmbH, TÜV Austria, Frauscher Sensonic, TRUMPF and the NVIDIA Corporation.

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

# Contents of the Appendix

# A    Generalisation of Signal Propagation

This section provides the derivations for the generalised signal propagation.

## A.1    Traditional Signal Propagation

One of the key assumptions in traditional signal propagation is that the weights are drawn from a zero-mean distribution with some variance $\sigma_w^2$. E.g. $w \sim \mathcal{N}(0, \sigma_w^2)$ or $w \sim \mathcal{U}\left(-\sqrt{3\sigma_w^2}, \sqrt{3\sigma_w^2}\right)$ are typical distributions for sampling initial weights. Also, bias parameters are typically assumed to be initialised with zeros, such that mean and variance are $\mu_b = 0$ and $\sigma_b^2 = 0$, respectively. Since the initial weights are drawn from i.i.d. samples, the first two (raw) moments of the pre-activations from eq. (1) be directly computed as follows:

$$\mathbb{E}[s_i] = \mathbb{E}\left[b_i + \sum_k w_{ik}\phi(s_k^-)\right]$$

$$= \mathbb{E}[b_i] + \sum_k \mathbb{E}[w_{ik}]\,\mathbb{E}[\phi(s_k^-)]$$

$$= \mu_b + \sum_k \mu_w\,\mathbb{E}[\phi(s_k^-)] = 0 \tag{13}$$

$$\mathbb{E}[s_i^2] = \mathbb{E}\left[\left(b_i + \sum_k w_{ik}\phi(s_k^-)\right)^2\right]$$

$$= \mathbb{E}[b_i^2] + 2\,\mathbb{E}[b_i]\sum_k \mathbb{E}[w_{ik}]\,\mathbb{E}[\phi(s_k^-)] + \mathbb{E}\left[\left(\sum_k w_{ik}\phi(s_k^-)\right)^2\right]$$

$$= (\sigma_b^2 + \mu_b^2) + 2\mu_b\sum_k \mu_w\,\mathbb{E}[\phi(s_k^-)] + \mathbb{E}\left[\sum_{k,k'} w_{ik}w_{ik'}\phi(s_k^-)\phi(s_{k'}^-)\right]$$

$$= \mathbb{E}\left[\sum_{k,k'} w_{ik}w_{ik'}\phi(s_k^-)\phi(s_{k'}^-)\right]$$

$$= \sum_k \mathbb{E}[w_{ik}^2]\,\mathbb{E}[\phi(s_k^-)^2] + \sum_{k,k'\neq k} \mathbb{E}[w_{ik}]\,\mathbb{E}[w_{ik'}]\,\mathbb{E}[\phi(s_k^-)\phi(s_{k'}^-)]$$

$$= \sum_k (\sigma_w^2 + \mu_w^2)\,\mathbb{E}[\phi(s_k^-)^2] + \sum_{k,k'\neq k} \mu_w^2\,\mathbb{E}[\phi(s_k^-)\phi(s_{k'}^-)]$$

$$= \underbrace{\sigma_w^2\sum_k \mathbb{E}[\phi(s_k^-)^2]}_{\text{Var}[s_i]}. \tag{14}$$

If we additionally assume that the pre-activations from the previous layer are identically distributed, such that

$$\forall k : \mathbb{E}[\phi(s_k^-)^2] = \mathbb{E}[\phi(s_1^-)^2], \tag{15}$$

we retrieve the traditional signal propagation formulas (equations 2 and 3):

$$\mathbb{E}[s_i] = 0$$

$$\mathbb{E}[s_i^2] = N\sigma_w^2\,\mathbb{E}[\phi(s_1^-)^2],$$

where $N$ is the number of incoming connections, such that $\boldsymbol{s}^- \in \mathbb{R}^N$.

For the first layer, the recursion formulas in this section can obviously not be used. The moments of the pre-activations in the first layer can be computed using a very similar derivation, however:

$$\mathbb{E}[s_i] = \mathbb{E}\left[b_i + \sum_k w_{ik}x_k\right] \qquad\qquad \mathbb{E}[s_i^2] = \mathbb{E}\left[\left(b_i + \sum_k w_{ik}x_k\right)^2\right]$$

$$= \mu_b + \mu_w\sum_k \mathbb{E}[x_k] = 0 \qquad\qquad = \sigma_w^2\sum_k \mathbb{E}[x_k^2].$$

There are two possible ways to interpret the inputs. If we wish to treat inputs as random variables, we would also need $x_k$ to be identically distributed to obtain

$$\mathbb{E}[s_i] = 0 \qquad\qquad \mathbb{E}[s_i^2] = N\sigma_w^2\,\mathbb{E}[x_1^2]$$

Alternatively, we can just use the input data as constants, such that $\mathbb{E}[x_k^2] = x_k^2$. In this scenario, the pre-activation moments are

$$\mathbb{E}[s_i] = 0 \qquad\qquad \mathbb{E}[s_i^2] = \sigma_w^2\|\boldsymbol{x}\|.$$

## A.2 Generalised Signal Propagation

In ICNNs, the zero-mean assumption on the weights does not hold. Therefore, we apply the signal propagation analysis assuming that weights have mean $\mu_w$ and variance $\sigma_w^2$. Furthermore, we also account for non-zero bias initialisation by not making any assumptions on the mean and variance of the bias parameters. In this setting, the first two (raw) moments of the pre-activations from eq. (1) are:

$$
\begin{aligned}
\mathbb{E}[s_i] &= \mathbb{E}\Big[b_i + \sum_k w_{ik}\phi(s_k^-)\Big] \\
&= \mu_b + \mu_w \sum_k \mathbb{E}[\phi(s_k^-)] 
\end{aligned}
\tag{16}
$$

$$
\begin{aligned}
\mathbb{E}[s_i^2] &= \mathbb{E}\Big[\Big(b_i + \sum_k w_{ik}\phi(s_k^-)\Big)^2\Big] \\
&= \sigma_b^2 + \mu_b^2 + 2\mu_b\mu_w \sum_k \mathbb{E}[\phi(s_k^-)] + \mathbb{E}\Big[\sum_{k,k'} w_{ik}w_{ik'}\phi(s_k^-)\phi(s_{k'}^-)\Big] \\
&= \sigma_b^2 + \mu_b^2 + 2\mu_b\mu_w \sum_k \mathbb{E}[\phi(s_k^-)] + (\sigma_w^2 + \mu_w^2)\sum_k \mathbb{E}[\phi(s_k^-)^2] + \mu_w^2 \sum_{k,k'\neq k} \mathbb{E}[\phi(s_k^-)\phi(s_{k'}^-)] \\
&= \sigma_b^2 + \mu_b^2 + 2\mu_b\mu_w \sum_k \mathbb{E}[\phi(s_k^-)] + \sigma_w^2 \sum_k \mathbb{E}[\phi(s_k^-)^2] + \mu_w^2 \sum_{k,k'} \mathbb{E}[\phi(s_k^-)\phi(s_{k'}^-)] \\
&= \sigma_b^2 + \sigma_w^2 \sum_k \mathbb{E}[\phi(s_k^-)^2] + \mu_w^2 \sum_{k,k'} \mathbb{E}[\phi(s_k^-)\phi(s_{k'}^-)] \\
&\quad - \mu_w^2 \sum_{k,k'} \mathbb{E}[\phi(s_k^-)]\,\mathbb{E}[\phi(s_{k'}^-)] + \Big(\mu_b + \mu_w \sum_k \mathbb{E}[\phi(s_k^-)]\Big)^2 \\
&= \underbrace{\sigma_b^2 + \sigma_w^2 \sum_k \mathbb{E}[\phi(s_k^-)^2] + \mu_w^2 \sum_{k,k'} \mathrm{Cov}[\phi(s_k^-), \phi(s_{k'}^-)]}_{\mathrm{Var}[s_i]} + \mathbb{E}[s_i]^2,
\end{aligned}
\tag{17}
$$

where

$$\mathrm{Cov}[\phi(s_k^-), \phi(s_{k'}^-)] = \mathbb{E}[\phi(s_k^-)\phi(s_{k'}^-)] - \mathbb{E}[\phi(s_k^-)]\,\mathbb{E}[\phi(s_{k'}^-)]$$

is the covariance.

Unlike in the traditional setting in section A.1, the pre-activations are not uncorrelated. This means that we also need to include the propagation of covariance in our generalised signal propagation

theory. We do this by considering the second mixed moment of the pre-activations:

$$\mathbb{E}_{i \neq j}[s_i s_j] = \mathbb{E}\left[\left(b_i + \sum_k w_{ik}\phi(s_k^-)\right)\left(b_j + \sum_k w_{jk}\phi(s_k^-)\right)\right]$$

$$= \mathbb{E}[b_i b_j] + 2\,\mathbb{E}\left[b_i \sum_k w_{jk}\phi(s_k^-)\right] + \mathbb{E}\left[\sum_{k,k'} w_{ik}w_{jk'}\phi(s_k^-)\phi(s_{k'}^-)\right]$$

$$= \mu_b^2 + 2\mu_b\mu_w \sum_k \mathbb{E}[\phi(s_k^-)] + \mu_w^2 \sum_{k,k'} \mathbb{E}[\phi(s_k^-)\phi(s_{k'}^-)]$$

$$= \left(\mu_b + \mu_w \sum_k \mathbb{E}[\phi(s_k^-)]\right)^2 + \mu_w^2 \sum_{k,k'} \mathrm{Cov}[\phi(s_k^-), \phi(s_{k'}^-)]$$

$$= \underbrace{\mu_w^2 \sum_{k,k'} \mathrm{Cov}[\phi(s_k^-), \phi(s_{k'}^-)]}_{\mathrm{Cov}_{i \neq j}[s_i, s_j]} + \mathbb{E}[s_i]\,\mathbb{E}[s_j]. \tag{18}$$

Note that this expression also appears in the variance formula (eq. 17).

We can again assume that pre-activations are (approximately) identically distributed to conclude that

$$\forall n \in \{1, 2\} : \forall k : \mathbb{E}\big[\phi(s_k^-)^n\big] = \mathbb{E}\big[\phi(s_1^-)^n\big].$$

Putting everything together, we arrive at our generalised signal propagation results from equations (5) and (6):

$$\mathbb{E}[s_i] = N\mu_w\,\mathbb{E}[\phi(s_1^-)] + \mu_b$$

$$\mathbb{E}[s_i s_j] = \delta_{ij}\left(N\sigma_w^2\,\mathbb{E}\big[\phi(s_1^-)^2\big] + \sigma_b^2\right) + \underbrace{\mu_w^2 \sum_{k,k'} \mathrm{Cov}[\phi(s_k^-), \phi(s_{k'}^-)] + \mathbb{E}[s_i]\,\mathbb{E}[s_j]}_{\mathrm{Cov}[s_i, s_j]}.$$

Here, we combined equations (17) and (18) into a single expression using the Kronecker delta, $\delta_{ij}$.

The identical distribution of the pre-activations also induces a particular structure in the covariance matrix. Because the diagonal entries represent the variance, they must have the same value. The off-diagonal entries happen to be the same — but typically different from the variance — as well. After all, eq. (18) is also independent of indices $i$ and $j$. As a result, the sum over entries in the covariance matrix can be written in terms of these two values:

$$\sum_{k,k'} \mathrm{Cov}[\phi(s_k^-), \phi(s_{k'}^-)] = \sum_k \left(\mathrm{Var}[\phi(s_k^-)] + \sum_{k' \neq k} \mathrm{Cov}[\phi(s_k^-), \phi(s_{k'}^-)]\right)$$

$$= N\big(\mathrm{Var}[\phi(s_1^-)] + (N-1)\,\mathrm{Cov}[\phi(s_1^-), \phi(s_2^-)]\big). \tag{19}$$

### A.3 Activation Function Kernels

Activation functions are not directly affected by the positive weights in ICNNs. Nevertheless, the propagation through non-linear activation functions also has to be reconsidered for our generalised theory. The main reason is the difference in covariance structure between the traditional and our generalised signal propagation.

When considering the propagation through non-linearities, the pre-activations are assumed to be Gaussian random variables. This is typically justified by the central limit theorem, which applies to sums of (weakly) independent random variables. In the traditional theory (sec. A.1), where $\mu_w = \mu_b = 0$, it can be verified that pre-activations are uncorrelated:

$$\mathbb{E}_{i \neq j}[s_i s_j] = \mathbb{E}\left[\left(b_i + \sum_k w_{ik}\phi(s_k^-)\right)\left(b_j + \sum_k w_{jk}\phi(s_k^-)\right)\right]$$

$$= \mathbb{E}[b_i b_j] + 2\,\mathbb{E}\left[b_i \sum_k w_{jk}\phi(s_k^-)\right] + \mathbb{E}\left[\sum_{k,k'} w_{ik}w_{jk'}\phi(s_k^-)\phi(s_{k'}^-)\right]$$

$$= \mu_b^2 + 2\mu_b\mu_w \sum_k \mathbb{E}[\phi(s_k^-)] + \mu_w^2 \sum_{k,k'} \mathbb{E}[\phi(s_k^-)\phi(s_{k'}^-)]$$

$$= 0. \tag{20}$$

Although this does not imply independence, it is often sufficient to claim weak independence. Also, empirical investigations typically confirm these assumptions (e.g. figure 1a).

In contrast, the results of our generalised theory (sec. A.2) suggest that pre-activations have non-zero correlation. Therefore, the Gaussian assumption on the pre-activations is hard to justify. Moreover, this assumption also clearly does not match empirical observations (e.g. figure 1c). Nevertheless, we adopt the Gaussian assumption for the generalised theory due to a lack of better options.

Consider some (non-linear) activation function, $\phi : \mathbb{R} \to \mathbb{R}$. The effects of $\phi$ on signal propagation can be captured by the matrix of mixed moments,

$$\mathbb{E}_{s_1,s_2 \sim \mathcal{N}(\mathbf{0},\boldsymbol{\Sigma})}[\phi(s_1)\phi(s_2)], \tag{21}$$

where $\boldsymbol{\Sigma} = \left(\begin{smallmatrix} 1 & \rho \\ \rho & 1 \end{smallmatrix}\right)\sigma^2$. Here, $\rho$ is the pair-wise correlation and $\sigma^2$ the variance of pre-activations. This matrix of mixed moments can also be used to define kernels (see e.g. Cho and Saul, 2009). Therefore, we will refer to this matrix as the *kernel* of the activation function. Note that for traditional signal propagation theory, only the diagonal is relevant.

For the sake of example, we show how to compute the kernel for the leaky ReLU function (Maas et al., 2013),

$$\mathrm{LReLU} : \mathbb{R} \to \mathbb{R} : x \mapsto \mathrm{LReLU}(x\,;\alpha) \begin{cases} x & x \geq 0 \\ \alpha x & x < 0 \end{cases}.$$

We choose the leaky ReLU function because it allows for analytical solutions. Throughout these computations, we assume pre-activations to have zero mean. Moreover, the covariance between any two features is assumed to be $\boldsymbol{\Sigma} = \left(\begin{smallmatrix} 1 & \rho \\ \rho & 1 \end{smallmatrix}\right)\sigma^2$. Here $\rho$ is the correlation and $\sigma^2$ is the variance of pre-activations.

Using the kernel for ReLU from (Daniely et al., 2016; Cho and Saul, 2009), the covariance for leaky ReLU is given by

$$\mathbb{E}[\,\mathrm{LReLU}(s_1\,;\alpha)\,\mathrm{LReLU}(s_2\,;\alpha)]$$

$$= \int_{-\infty}^{0}\int_{-\infty}^{0} \alpha^2 s_1 s_2\, p_{\mathcal{N}}(s_1,s_2\,;\mathbf{0},\boldsymbol{\Sigma})\,\mathrm{d}s_1\,\mathrm{d}s_2 + \int_{-\infty}^{0}\int_{0}^{\infty} \alpha s_1 s_2\, p_{\mathcal{N}}(s_1,s_2\,;\mathbf{0},\boldsymbol{\Sigma})\,\mathrm{d}s_1\,\mathrm{d}s_2$$

$$+ \int_{0}^{\infty}\int_{-\infty}^{0} \alpha s_1 s_2\, p_{\mathcal{N}}(s_1,s_2\,;\mathbf{0},\boldsymbol{\Sigma})\,\mathrm{d}s_1\,\mathrm{d}s_2 + \int_{0}^{\infty}\int_{0}^{\infty} s_1 s_2\, p_{\mathcal{N}}(s_1,s_2\,;\mathbf{0},\boldsymbol{\Sigma})\,\mathrm{d}s_1\,\mathrm{d}s_2$$

$$= (1+\alpha^2)\,\mathbb{E}_{\mathcal{N}(\mathbf{0},\boldsymbol{\Sigma})}[\mathrm{ReLU}(s_1)\,\mathrm{ReLU}(s_2)] - 2\alpha\,\mathbb{E}_{\mathcal{N}(\mathbf{0},\bar{\boldsymbol{\Sigma}})}[\mathrm{ReLU}(s_1)\,\mathrm{ReLU}(s_2)]$$

$$= (1+\alpha^2)\frac{\sigma^2}{2\pi}\Big(\sqrt{1-\rho^2} + \rho\arccos(-\rho)\Big) - 2\alpha\frac{\sigma^2}{2\pi}\Big(\sqrt{1-\rho^2} - \rho\arccos(\rho)\Big) \tag{22}$$

$$= (1+\alpha^2)\frac{\sigma^2}{2\pi}\Big(\sqrt{1-\rho^2} + \rho\arccos(-\rho)\Big) - 2\alpha\frac{\sigma^2}{2\pi}\Big(\sqrt{1-\rho^2} - \rho\pi + \rho\arccos(-\rho)\Big)$$

$$= (1-\alpha)^2\,\frac{\sigma^2}{2\pi}\Big(\sqrt{1-\rho^2} + \rho\arccos(-\rho)\Big) + \alpha\sigma^2\rho, \tag{23}$$

where $\bar{\boldsymbol{\Sigma}} = \left(\begin{smallmatrix} 1 & -\rho \\ -\rho & 1 \end{smallmatrix}\right)\sigma^2$ appears as a side-effect of flipping the integration boundaries. Note that this expression also captures the squared mean ($\rho = 0$) and second raw moment ($\rho = 1$).

### A.4 Generalised Backward Propagation

The generalisation to non-zero mean also applies to backpropagation, where signals are the so-called deltas, the derivatives of the loss, $L$, w.r.t. the pre-activations. These deltas can be defined using the $\mathbb{R}^M \to \mathbb{R}^N$ mapping:

$$\boldsymbol{\delta}^- = \frac{\partial L}{\partial \boldsymbol{s}^-} = \phi'(\boldsymbol{s}^-)\boldsymbol{W}^\mathsf{T}\boldsymbol{\delta}. \tag{24}$$

Here, $\boldsymbol{W}$ are the weights of the layer that takes $\phi(\boldsymbol{s}^-)$ as inputs. Note that

The first two (raw) moments of the deltas from eq. (24) are:

$$\mathbb{E}[\delta_j^-] = \mathbb{E}\Big[\phi'(s_j^-)\sum_k \delta_k w_{kj}\Big]$$

$$= \mu_w\,\mathbb{E}[\phi'(s_j^-)]\sum_k \mathbb{E}[\delta_k] \tag{25}$$

$$\mathbb{E}\big[(\delta_j^-)^2\big] = \mathbb{E}\Big[\Big(\phi'(s_j^-)\sum_k \delta_k w_{kj}\Big)^2\Big]$$

$$= \mathbb{E}\Big[\phi'(s_j^-)^2\sum_{k,k'}\delta_k\delta_{k'}w_{kj}w_{k'j}\Big]$$

$$= (\sigma_w^2 + \mu_w^2)\,\mathbb{E}\big[\phi'(s_j^-)^2\big]\sum_k \mathbb{E}\big[\delta_k^2\big] + \mu_w^2\,\mathbb{E}\big[\phi'(s_j^-)^2\big]\sum_{k,k'\neq k}\mathbb{E}[\delta_k\delta_{k'}]$$

$$= \sigma_w^2\,\mathbb{E}\big[\phi'(s_j^-)^2\big]\sum_k \mathbb{E}\big[\delta_k^2\big] + \mu_w^2\,\mathbb{E}\big[\phi'(s_j^-)^2\big]\sum_{k,k'}\mathbb{E}[\delta_k\delta_{k'}]. \tag{26}$$

The second mixed moment can be derived in a similar way:

$$\mathbb{E}_{i\neq j}[\delta_i^-\delta_j^-] = \mathbb{E}\Big[\Big(\phi'(s_i^-)\sum_k \delta_k w_{ki}\Big)\Big(\phi'(s_j^-)\sum_k \delta_k w_{kj}\Big)\Big]$$

$$= \mathbb{E}\Big[\phi'(s_i^-)\phi'(s_j^-)\sum_{k,k'}w_{ki}w_{k'j}\delta_k\delta_{k'}\Big]$$

$$= \mu_w^2\,\mathbb{E}[\phi'(s_i^-)\phi'(s_j^-)]\sum_{k,k'}\mathbb{E}[\delta_k\delta_{k'}]. \tag{27}$$

From the forward pass (see section A.2), we know that the moments of pre-activations are independent of their index, i.e. $\forall k : \mathbb{E}\big[(s_k^-)^n\big] = \mathbb{E}\big[(s_1^-)^n\big]$. As a result, we also have $\mathbb{E}\big[\phi'(s_k^-)^n\big] = \mathbb{E}\big[\phi'(s_1^-)^n\big]$ and therefore

$$\forall n \in \{1,2\} : \forall k : \mathbb{E}\big[(\delta_k^-)^n\big] = \mathbb{E}\big[(\delta_1^-)^n\big].$$

Putting everything together, we obtain the generalised signal propagation for the deltas during backpropagation:

$$\mathbb{E}[\delta_j^-] = M\mu_w\,\mathbb{E}[\phi'(s_1^-)]\,\mathbb{E}[\delta_1]$$

$$\mathbb{E}[\delta_i^-\delta_j^-] = \delta_{ij}\,M\sigma_w^2\,\mathbb{E}\big[\phi'(s_1^-)^2\big]\,\mathbb{E}\big[\delta_1^2\big] + \mu_w^2\,\mathbb{E}[\phi'(s_i^-)\phi'(s_j^-)]\sum_{k,k'}\mathbb{E}[\delta_k\delta_{k'}],$$

where $\delta_{ij}$ is the Kronecker delta.

The covariance for the derivative of LReLU under the same assumptions as in section A.3 would be given by

$$\mathbb{E}[\,\mathrm{LReLU}'(s_1\,;\alpha)\,\mathrm{LReLU}'(s_2\,;\alpha)]$$

$$= \int_{-\infty}^0\int_{-\infty}^0 \alpha^2 p_\mathcal{N}(s_1,s_2\,;\mathbf{0},\boldsymbol{\Sigma})\,\mathrm{d}s_1\,\mathrm{d}s_2 + \int_{-\infty}^0\int_0^\infty \alpha\,p_\mathcal{N}(s_1,s_2\,;\mathbf{0},\boldsymbol{\Sigma})\,\mathrm{d}s_1\,\mathrm{d}s_2$$

$$\quad + \int_0^\infty\int_{-\infty}^0 \alpha\,p_\mathcal{N}(s_1,s_2\,;\mathbf{0},\boldsymbol{\Sigma})\,\mathrm{d}s_1\,\mathrm{d}s_2 + \int_0^\infty\int_0^\infty p_\mathcal{N}(s_1,s_2\,;\mathbf{0},\boldsymbol{\Sigma})\,\mathrm{d}s_1\,\mathrm{d}s_2$$

$$= (1+\alpha^2)\frac{1}{2\pi}\arccos(-\rho) + 2\alpha\frac{1}{2\pi}\arccos(\rho)$$

$$= (1+\alpha^2)\frac{1}{2\pi}\arccos(-\rho) + 2\alpha\frac{1}{2\pi}\big(\pi - \arccos(-\rho)\big)$$

$$= (1-\alpha)^2\frac{1}{2\pi}\arccos(-\rho) + \alpha \tag{28}$$

## A.5 Convolutional Layers

Our generalised signal propagation (see section A.2) is also applicable to convolutional layers. This section provides the derivations for a one-dimensional cross-correlation,

$$s_{ia} = b_i + \sum_{c,k} w_{ick}\phi(s^-_{c(a+k)}),$$

but can be generalised in a similar way to higher dimensional operations. The first two moments (cf. equations 16 and 17) are given by

$$\mathbb{E}[s_{ia}] = \mathbb{E}\Big[b_i + \sum_{c,k} w_{ick}\phi\big(s^-_{c(a+k)}\big)\Big]$$

$$= \mu_b + \mu_w \sum_{c,k} \mathbb{E}\big[\phi\big(s^-_{c(a+k)}\big)\big] \tag{29}$$

$$\mathbb{E}\big[s^2_{ia}\big] = \mathbb{E}\bigg[\Big(b_i + \sum_{c,k} w_{ick}\phi\big(s^-_{c(a+k)}\big)\Big)^2\bigg]$$

$$= \sigma^2_b + \mu^2_b + 2\mu_b\mu_w \sum_{c,k} \mathbb{E}\big[\phi\big(s^-_{c(a+k)}\big)\big] + \sigma^2_w \sum_{c,k} \mathbb{E}\big[\phi\big(s^-_{c(a+k)}\big)^2\big] + \mu^2_w \sum_{c,k}\sum_{c',k'} \mathbb{E}\big[\phi\big(s^-_{c(a+k)}\big)\phi\big(s^-_{c'(a+k')}\big)\big]$$

$$= \underbrace{\sigma^2_b + \sigma^2_w \sum_{c,k} \mathbb{E}\big[\phi\big(s^-_{c(a+k)}\big)^2\big] + \mu^2_w \sum_{c,k}\sum_{c',k'} \mathrm{Cov}\big[\phi\big(s^-_{c(a+k)}\big),\phi\big(s^-_{c'(a+k')}\big)\big]}_{\mathrm{Var}[s_{ia}]} + \mathbb{E}[s_{ia}]^2.$$

$$\tag{30}$$

For the second mixed moment we find (cf. eq 18)

$$\mathbb{E}_{i\neq j}\big[s_{ia}s_{j(a+d)}\big] = \mathbb{E}\bigg[\Big(b_i + \sum_{c,k} w_{ick}\phi\big(s^-_{c(a+k)}\big)\Big)\Big(b_j + \sum_{c,k} w_{jck}\phi\big(s^-_{c(a+d+k)}\big)\Big)\bigg]$$

$$= \mu^2_b + \mu_b\mu_w \sum_{c,k}\Big(\mathbb{E}\big[\phi\big(s^-_{c(a+k)}\big)\big] + \mathbb{E}\big[\phi\big(s^-_{c(a+d+k)}\big)\big]\Big)$$

$$+ \mu^2_w \sum_{c,k}\sum_{c',k'} \mathbb{E}\big[\phi\big(s^-_{c(a+k)}\big)\phi\big(s^-_{c'(a+d+k')}\big)\big]$$

$$= \underbrace{\mu^2_w \sum_{c,k}\sum_{c',k'} \mathrm{Cov}\big[\phi\big(s^-_{c(a+k)}\big),\phi\big(s^-_{c'(a+d+k')}\big)\big]}_{\mathrm{Cov}[s_{ia},s_{j(a+d)}]} + \mathbb{E}[s_{ia}]\,\mathbb{E}[s_{j(a+d)}]. \tag{31}$$

However, this mixed moment does not capture the covariance between elements in the same channel, but on different positions. Therefore, we additionally include the following result:

$$\mathbb{E}_{d\neq 0}\big[s_{ia}s_{i(a+d)}\big] = \mathbb{E}\bigg[\Big(b_i + \sum_{c,k} w_{ick}\phi\big(s^-_{c(a+k)}\big)\Big)\Big(b_i + \sum_{c,k} w_{ick}\phi\big(s^-_{c(a+d+k)}\big)\Big)\bigg]$$

$$= \sigma^2_b + \mu^2_b + \mu_b\mu_w \sum_{c,k}\Big(\mathbb{E}\big[\phi\big(s^-_{c(a+k)}\big)\big] + \mathbb{E}\big[\phi\big(s^-_{c(a+d+k)}\big)\big]\Big)$$

$$+ \sigma^2_w \sum_{c,k} \mathbb{E}\big[\phi\big(s^-_{c(a+k)}\big)\phi\big(s^-_{c(a+d+k)}\big)\big] + \mu^2_w \sum_{c,k}\sum_{c',k'} \mathbb{E}\big[\phi\big(s^-_{c(a+k)}\big)\phi\big(s^-_{c'(a+d+k')}\big)\big]$$

$$= \sigma^2_b + \sigma^2_w \sum_{c,k} \mathbb{E}\big[\phi\big(s^-_{c(a+k)}\big)\phi\big(s^-_{c(a+d+k)}\big)\big]$$

$$+ \mu^2_w \sum_{c,k}\sum_{c',k'} \mathrm{Cov}\big[\phi\big(s^-_{c(a+k)}\big),\phi\big(s^-_{c'(a+d+k')}\big)\big] + \mathbb{E}[s_{ia}]\,\mathbb{E}[s_{i(a+d)}] \tag{32}$$

Putting these results together, we can summarise the signal propagation through a convolutional layer as follows:

$$\mathbb{E}[s_{ia}] = \mu_b + \mu_w \sum_{c,k} \mathbb{E}\big[\phi\big(s_{c(a+k)}^-\big)\big]$$

$$\mathbb{E}[s_{ia}s_{j(a+d)}] = \delta_{ij}\left(\sigma_b^2 + \sigma_w^2 \sum_{c,k} \mathbb{E}\big[\phi\big(s_{c(a+k)}^-\big)\phi\big(s_{c(a+d+k)}^-\big)\big]\right)$$
$$+ \mu_w^2 \sum_{c,k}\sum_{c',k'} \mathrm{Cov}\big[\phi\big(s_{c(a+k)}^-\big), \phi\big(s_{c'(a+d+k')}^-\big)\big] + \mathbb{E}[s_{ia}]\,\mathbb{E}[s_{j(a+d)}]$$

Here, we use the Kronecker delta to combine equations (30), (31) and (32) into one equation.

## B    Deriving the Initialisation

This section aims to show how the ICNN initialisation can be derived from our generalised signal propagation theory.

### B.1    Zero Mean Pre-Activations

The goal of a good initialisation is to have pre-activations with similar statistics in every layer. In the traditional theory, mean and covariance are zero in every layer by default. Therefore, variance is the only statistic that needs to be controlled. This is why traditional initialisation methods (LeCun et al., 1998; Glorot and Bengio, 2010; He et al., 2015) typically focus on the standard deviation of weights.

Consider the equation for pre-activations from eq. (1). Assuming $\phi = \mathrm{LReLU}$, we can use the kernel from eq. (23). Plugging the result for $\rho = 1$ into eq. (3) we obtain the traditional variance propagation

$$\mathbb{E}\big[s_i^2\big] = N\sigma_w^2(1+\alpha^2)\frac{1}{2}\mathbb{E}\big[(s_1^-)^2\big].$$

By requiring that $\mathbb{E}\big[s_i^2\big] = \mathbb{E}\big[(s_1^-)^2\big] = \sigma_*^2 > 0$, we obtain the fixed point equation

$$\sigma_*^2 = N\sigma_w^2(1+\alpha^2)\frac{1}{2}\sigma_*^2,$$

which can be solved for $\sigma_w^2$ to obtain the initialisation variance for (regular) LReLU networks:

$$\sigma_w^2 = \frac{2}{1+\alpha^2}\frac{1}{N}.$$

Here, $2/(1+\alpha^2)$ is sometimes also referred to as the *gain* of the initialisation (cf. Saxe et al., 2014).

We adopt a similar approach for our generalised propagation results. In contrast to the traditional setting, however, we can not directly rely on the activation kernels. After all, the pre-activations in our general setting are not automatically zero.

**Derivation of Bias Mean**    By setting eq. (5) to zero and solving for $\mu_b$, we get

$$\mu_b = -N\mu_w\,\mathbb{E}[\phi(s_1^-)].$$

This also makes it easier to use the activation kernels, which assume zero mean inputs. Note, however, that the activation kernels theoretically do not apply to our setting (see sec. A.3). Nevertheless, we assume that the activation kernels approximate the actual dynamics well enough. Plugging in the square root of the LReLU kernel (eq. 23) with $\rho = 0$, we obtain the bias mean for $\phi = \mathrm{LReLU}$:

$$\mu_b = -N\mu_w(1-\alpha)\sqrt{\frac{1}{2\pi}\Big(\mathbb{E}\big[(s_1^-)^2\big] - \mathbb{E}[s_1^-]^2\Big)}$$
$$= -N\mu_w(1-\alpha)\sqrt{\frac{1}{2\pi}\mathbb{E}\big[(s_1^-)^2\big]}. \tag{33}$$

## B.2 Variance and Covariance Fixed Points

To obtain fixed points for the variance and covariance, we start from equations (17) and (18), respectively. Assuming identically distributed pre-activations, we can use equations (15) and (19), to get expressions without sums:

$$\mathbb{E}\big[s_i^2\big] = \sigma_b^2 + \sigma_w^2 N\,\mathbb{E}\big[\phi(s_1^-)^2\big] + \mathbb{E}_{i\neq j}[s_i s_j]$$

$$\mathbb{E}_{i\neq j}[s_i s_j] = \mu_w^2 N\Big(\,\mathrm{Var}\big[\phi(s_1^-)^2\big] + (N-1)\,\mathrm{Cov}\big[\phi(s_1^-), \phi(s_2^-)\big]\Big)$$

For $\phi = \mathrm{LReLU}$, we can use the kernel from eq. (23) with $\rho = 1$ to work out the moments for the variance,

$$\mathbb{E}\big[s_i^2\big] = \sigma_b^2 + \sigma_w^2 N(1+\alpha^2)\frac{1}{2}\,\mathbb{E}\big[(s_1^-)^2\big] + \mathbb{E}_{i\neq j}[s_i s_j], \tag{34}$$

and with $\rho = c^- = \mathrm{Corr}[s_1^-, s_2^-] = \mathbb{E}[s_1^- s_2^-]/\mathbb{E}\big[(s_1^-)^2\big]$ to obtain the covariance

$$\mathbb{E}_{i\neq j}[s_i s_j] = \mu_w^2 N\Big(\,\mathbb{E}\big[\mathrm{LReLU}(s_1^-)^2\big] + (N-1)\,\mathbb{E}\big[\mathrm{LReLU}(s_1^-)\,\mathrm{LReLU}(s_2^-)\big] - N\,\mathbb{E}\big[\mathrm{LReLU}(s_1^-)\big]^2\Big)$$

$$= \mu_w^2 N\Bigg(\frac{1+\alpha^2}{2}\,\mathbb{E}\big[(s_1^-)^2\big] - N\frac{(1-\alpha)^2}{2\pi}\,\mathbb{E}\big[(s_1^-)^2\big]$$

$$+ (N-1)\bigg(\frac{(1-\alpha)^2}{2\pi}\,\mathbb{E}\big[(s_1^-)^2\big]\Big(\sqrt{1-(c^-)^2} + c^-\arccos(-c^-)\Big) + \alpha\,\mathbb{E}\big[(s_1^-)^2\big]c^-\bigg)\Bigg)$$

$$= \mu_w^2 \frac{N}{2\pi}\,\mathbb{E}\big[(s_1^-)^2\big]\bigg((1+\alpha^2)\pi - N(1-\alpha)^2$$

$$+ (N-1)\Big((1-\alpha)^2\Big(\sqrt{1-(c^-)^2} + c^-\arccos(-c^-)\Big) + 2\pi\alpha c^-\Big)\bigg)$$

For the sake of readability, we introduce $f_{\mathrm{c}} : [-1,1] \to \mathbb{R}$ to denote

$$f_{\mathrm{c}}(\rho) = \frac{N}{2\pi}\bigg((1+\alpha^2)\pi - N(1-\alpha)^2 + (N-1)\Big((1-\alpha)^2\Big(\sqrt{1-\rho^2} + \rho\arccos(-\rho)\Big) + 2\pi\alpha\rho\Big)\bigg), \tag{35}$$

such that the covariance can be written more compactly as

$$\mathbb{E}_{i\neq j}[s_i s_j] = \mu_w^2\,\mathbb{E}\big[(s_1^-)^2\big]\,f_{\mathrm{c}}\big(\mathrm{Corr}[s_1^-, s_2^-]\big) \tag{36}$$

**Derivation of Bias and Weight Variances**   The fixed point equation for the variance (eq. 34) with $\mathbb{E}\big[s_i^2\big] = \mathbb{E}\big[(s_1^-)^2\big] = \sigma_*^2$ can be written as

$$\sigma_*^2 = \sigma_b^2 + \sigma_w^2 N(1+\alpha^2)\frac{1}{2}\sigma_*^2 + \sigma_*^2\mu_w^2 f_{\mathrm{c}}\big(\mathrm{Corr}[s_1^-, s_2^-]\big),$$

where $f_{\mathrm{c}}$ is taken from equation (35). Note that we retrieve eq. (10) in the main text from this result by setting $\alpha = 0$ and replacing $\mu_w^2 f_{\mathrm{c}}\big(\mathrm{Corr}[s_1^-, s_2^-]\big)$ by the correlation fixed point, $\rho_*$. We elaborate on the latter when discussing the fixed point equation for the correlation. Solving the fixed point equation for $\sigma_w^2$, we find

$$\sigma_w^2 = \frac{2}{1+\alpha^2}\frac{1}{N}\bigg(1 - \frac{\sigma_b^2}{\sigma_*^2} - \mu_w^2 f_{\mathrm{c}}\big(\mathrm{Corr}[s_1^-, s_2^-]\big)\bigg). \tag{37}$$

By initialising the bias vector with a constant value, such that $\sigma_b^2 = 0$, the expression above becomes independent of the fixed point $\sigma_*^2$:

$$\sigma_w^2 = \frac{2}{1+\alpha^2}\frac{1}{N}\bigg(1 - \mu_w^2 f_{\mathrm{c}}\big(\mathrm{Corr}[s_1^-, s_2^-]\big)\bigg). \tag{38}$$

Using similar substitutions as before, we obtain the result from the main text (eq. 11). A further simplification is possible by revisiting eq. (36) and observing that if $\mathbb{E}\big[s_i^2\big] = \mathbb{E}\big[(s_1^-)^2\big]$, we have $\mathrm{Corr}[s_i, s_j] = \mu_w^2 f_{\mathrm{c}}\big(\mathrm{Corr}[s_1^-, s_2^-]\big)$, such that

$$\sigma_w^2 = \frac{2}{1+\alpha^2}\frac{1}{N}\bigg(1 - \mathrm{Corr}[s_i, s_j]\bigg). \tag{39}$$

**Derivation of Weight Mean** Because eq. (36) is mainly a function of correlation, we consider a fixed point equation in terms of correlation rather than covariance. Concretely, we set $\text{Corr}[s_i, s_j] = \text{Corr}[s_1^-, s_2^-] = \rho_*$ to obtain the fixed point equation

$$\rho_* = \mu_w^2 \frac{\mathbb{E}\big[(s_1^-)^2\big]}{\mathbb{E}\big[s_1^2\big]} f_{\text{c}}(\rho_*),$$

where we used the identical distribution assumption as follows

$$\text{Corr}[s_i, s_j] = \frac{\mathbb{E}_{i \neq j}\big[s_i s_j\big]}{\sqrt{\mathbb{E}\big[s_i^2\big] \mathbb{E}\big[s_j^2\big]}} = \frac{\mathbb{E}_{i \neq j}\big[s_i s_j\big]}{\mathbb{E}\big[s_1^2\big]}.$$

Note that for $\alpha = 0$ and $\mathbb{E}\big[s_1^2\big] = \mathbb{E}\big[(s_1^-)^2\big]$, this fixed point equation corresponds to eq. (9) in the main text. Solving the fixed point equation for $\mu_w^2$, we find

$$\mu_w^2 = \frac{\mathbb{E}\big[s_1^2\big]}{\mathbb{E}\big[(s_1^-)^2\big]} \rho_* f_{\text{c}}(\rho_*)^{-1}. \tag{40}$$

Under similar conditions as previously stated, this gives us the result from the main text (eq. 12). We can additionally plug in eq. (34) into this result to get rid of $\mathbb{E}\big[s_1^2\big]$:

$$\mu_w^2 = \left( \frac{\sigma_b^2}{\mathbb{E}\big[(s_1^-)^2\big]} + \sigma_w^2 N \frac{1 + \alpha^2}{2} + \rho_* \right) \rho_* f_{\text{c}}(\rho_*)^{-1}.$$

Assuming $\sigma_b^2 = 0$ again, we obtain an expression that is independent of $\mathbb{E}\big[(s_1^-)^2\big]$:

$$\mu_w^2 = \left( \sigma_w^2 N \frac{1 + \alpha^2}{2} + \rho_* \right) \rho_* f_{\text{c}}(\rho_*)^{-1}. \tag{41}$$

## B.3 Initialisation Parameters

Putting everything together from sections B.1 and B.2, we obtain an initialisation for fully-connected layers with features produced by a LReLU non-linearity. The initialisation parameters are (see equations 33, 38 and 41):

$$\mu_w = \pm \sqrt{\left( \sigma_w^2 N \frac{1 + \alpha^2}{2} + \rho_* \right) \rho_* f_{\text{c}}(\rho_*)^{-1}} \qquad \sigma_w^2 = \frac{2}{1 + \alpha^2} \frac{1}{N} \left( 1 - \mu_w^2 f_{\text{c}}\big(\text{Corr}[s_1^-, s_2^-]\big) \right)$$

$$\mu_b = -N\mu_w(1 - \alpha)\sqrt{\frac{1}{2\pi} \mathbb{E}\big[(s_1^-)^2\big]} \qquad \sigma_b^2 = 0.$$

Note that the mean for the weights is the only result that directly depends on the fixed point. However, because these results are strongly connected, they have to be considered simultaneously. Practically, this means that we have to focus on the joint fixed point $(\sigma_*, \rho_*)$ instead of the individual parts. Considering equations (39) and (40), it is clear that the joint fixed point can be obtained with the following formulations

$$\mu_w = \pm \sqrt{\rho_* f_{\text{c}}(\rho_*)^{-1}} \qquad\qquad \sigma_w^2 = \frac{2}{1 + \alpha^2} \frac{1}{N}(1 - \rho_*) \tag{42}$$

$$\mu_b = -N\mu_w(1 - \alpha)\sqrt{\frac{\sigma_*^2}{2\pi}} \qquad\qquad \sigma_b^2 = 0. \tag{43}$$

Our initialisation parameters still depend on the choice of the joint fixed point $\sigma_*^2$ and $\rho_*$. Normally, $\sigma_*^2 = 1$ is assumed because that is how the input data is typically normalised. In general, we can assume $\sigma_*^2$ to be the variance of the input data. The ideal scenario for correlation is to have $\rho_* = 0$, i.e. uncorrelated features. However, when plugging this into our initialisation parameters, we retrieve the traditional setting from section A.1, where $\mu_w = 0$. In other words, uncorrelated features are only possible if $\mu_w = 0$. Also, $|\rho_*| < 1$ is desirable, because $\rho = 1$ practically corresponds to all

values being the same. Therefore, we (arbitrarily) choose to set $\rho_* = \frac{1}{2}$. One argument in favour of this choice is that eq. (35) has a relatively nice solution

$$f_c\left(\frac{1}{2}\right) = \frac{N}{2\pi}\left((1+\alpha^2)\pi - N(1-\alpha)^2 + (N-1)\left((1-\alpha)^2\left(\frac{\sqrt{3}}{2} + \frac{\pi}{3}\right) + \pi\alpha\right)\right)$$

$$= \frac{N}{12\pi}\left(6(1+\alpha^2)\pi - 6N(1-\alpha)^2 + (N-1)\left((1-\alpha)^2(3\sqrt{3}+2\pi) + 6\pi\alpha\right)\right)$$

$$= \frac{N}{12\pi}\left(6(1-\alpha)^2\pi + 12\alpha\pi - 6N(1-\alpha)^2 + (N-1)(1-\alpha)^2(3\sqrt{3}+2\pi) + 6(N-1)\pi\alpha\right)$$

$$= \frac{N}{12\pi}\left((1-\alpha)^2\left(6\pi - 6N + (N-1)(3\sqrt{3}+2\pi)\right) + 6(N+1)\pi\alpha\right)$$

$$= \frac{N}{12\pi}\left((1-\alpha)^2\left(6(\pi-1) + (N-1)(3\sqrt{3}+2\pi-6)\right) + 6(N+1)\pi\alpha\right).$$

Filling out the fixed point $(\sigma_*^2, \rho_*) = (1, \frac{1}{2})$, we obtain the final initialisation parameters for ICNNs with LReLU activation functions:

$$\mu_w = \pm\sqrt{\frac{6\pi}{N\left((1-\alpha)^2\left(6\pi - 6N + (N-1)(3\sqrt{3}+2\pi)\right) + 6(N+1)\pi\alpha\right)}} \qquad \sigma_w^2 = \frac{1}{1+\alpha^2}\frac{1}{N}$$

$$\mu_b = \mp\sqrt{\frac{3N(1-\alpha)^2}{(1-\alpha)^2\left(6\pi - 6N + (N-1)(3\sqrt{3}+2\pi)\right) + 6(N+1)\pi\alpha}} \qquad \sigma_b^2 = 0.$$

Note again that these results correspond to the results in the main paper where $\alpha = 0$.

### B.4  Fixed Point Stability

The goal of this section is to establish the stability of the joint fixed point we used for our initialisation from section B.3. Therefore, we analyse the Jacobian of the mapping

$$f_* : \mathbb{R}^+ \times [-1,1] \to \mathbb{R}^+ \times [-1,1] : (\sigma^2, \rho) \mapsto \left(\sigma_b^2 + \sigma_w^2 N(1+\alpha^2)\frac{1}{2}\sigma^2 + \sigma^2\mu_w^2 f_c(\rho), \mu_w^2 f_c(\rho)\right),$$

which combines equations (34) and (36) with $f_c$ from eq. (35). Filling out our initialisation parameters from equations (42) and (43), we obtain

$$f_*(\sigma^2, \rho) = \left(1 - \rho_* + \sigma^2\rho_*\frac{f_c(\rho)}{f_c(\rho_*)}, \rho_*\frac{f_c(\rho)}{f_c(\rho_*)}\right). \tag{44}$$

The stability of $f_*$ can be analysed by computing its Jacobian

$$\boldsymbol{J}_{f_*}(\sigma^2, \rho) = \begin{bmatrix} \rho_*\frac{f_c(\rho)}{f_c(\rho_*)} & \sigma^2\rho_*\frac{f_c'(\rho)}{f_c(\rho_*)} \\ 0 & \rho_*\frac{f_c'(\rho)}{f_c(\rho_*)} \end{bmatrix},$$

where

$$f_c'(\rho) = \frac{N}{2\pi}(N-1)(1-\alpha)^2\frac{\partial}{\partial\rho}\left(\sqrt{1-\rho^2} + \rho\arccos(-\rho)\right) + \frac{N}{2\pi}(N-1)2\pi\alpha$$

$$= \frac{N}{2\pi}(N-1)(1-\alpha)^2\left(\frac{-\rho}{\sqrt{1-\rho^2}} + \arccos(-\rho) + \rho\frac{1}{\sqrt{1-\rho^2}}\right) + N(N-1)\alpha$$

$$= \frac{N}{2\pi}(N-1)(1-\alpha)^2\arccos(-\rho) + N(N-1)\alpha.$$

The eigenvalues of the Jacobian are the roots of the characteristic polynomial

$$|\lambda\boldsymbol{I} - \boldsymbol{J}_{f_*}(\sigma^2, \rho)| = \left(\lambda - \rho_*\frac{f_c(\rho)}{f_c(\rho_*)}\right)\left(\lambda - \rho_*\frac{f_c'(\rho)}{f_c(\rho_*)}\right),$$

which happen to be the entries on the diagonal:

$$\lambda_1 = \rho_* \frac{f_c(\rho)}{f_c(\rho_*)} \qquad\qquad \lambda_2 = \rho_* \frac{f_c'(\rho)}{f_c(\rho_*)}.$$

Evaluating the eigenvalues of the Jacobian at the joint fixed point, $(\sigma_*^2, \rho_*)$, we find

$$\lambda_1 = \rho_* \qquad\qquad \lambda_2 = \rho_* \frac{f_c'(\rho_*)}{f_c(\rho_*)}.$$

Setting $\rho_* = \frac{1}{2}$ and $\alpha = 0$, this becomes

$$\lambda_2 = \frac{(N-1)2\pi}{6\pi - 6 + (N-1)(3\sqrt{3} + 2\pi - 6)},$$

which is only less than one if

$$N < 1 + \frac{2\pi - 2}{2 - \sqrt{3}} \approx 17.$$

As a result, the fixed point $(\sigma_*^2, \rho_*) = (1, \frac{1}{2})$ is not stable in practical settings.

## C   Additional Experiments

This section presents experimental details and results that did not make it into the main paper.

### C.1   Computing resources and budget

For our experiments, we had access to server infrastructure with multiple GPUs. We mainly used NVIDIA Titan V and NVIDIA RTX 2080Ti graphics cards with approximately 12 GB of memory for the computer vision experiments. Experiments were run in parallel on up to 8 GPUs. To maximise GPU usage, repetitions also ran in parallel on single cards. The Tox21 experiments were run on a desktop PC with one NVIDIA GTX 1070Ti, which has 8 GB of on-board memory.

A single run for the experiments in Fig. 2, Fig. 5, and Fig. 10 took approximately one minute and used no more than 2 GB of GPU memory on MNIST. For CIFAR10 and CIFAR100, a single run took up to four minutes and used up to 4 GB of GPU memory. As a result, an (over-)estimate for the compute time for Figure 2 is 6 hours or 18 hours for Figure 5. All the repetitions for the Tox21 results in Table 1, were obtained in 2 hours — i.e. 30 minutes per model — using a little over 500 MB of GPU memory.

The hyper-parameter search that lead to Table 3, required the most compute. For these estimates, we processed the timestamps for each run in our logs. Since we did not log memory consumption, we can not reliably report this data. The MNIST non-convex baseline model search required approximately 24 hours with a median run-time of less than four minutes. The search for the non-convex model on CIFAR10 required 9 hours with a median run-time of approximately two minutes. Note that five runs ran simultaneously (on the same card) for MNIST, but only three runs ran at the same time for CIFAR10. The search for "ICNN" took 11 hours for MNIST and 44 hours for CIFAR10. The search for "ICNN + skip" took 14 hours on both MNIST and CIFAR10. The search for "ICNN + init" took 45 hours on MNIST and 40 hours for CIFAR10. This gives a total of 201 hours wall-clock computation time for the hyper-parameter search. Training the resulting ten repetitions for the eight resulting models required an additional 2.5 hours for MNIST and 3.5 hours for CIFAR10 to obtain figure 3.

### C.2   Computer Vision Benchmarking Datasets

In this section, we provide additional details on the experiments on computer vision datasets. Figure 5 extends Figure 2 from the main paper with learning curves for networks with three and seven hidden layers. Table 3 lists the best hyper-parameters for the search behind Figure 3. Table 2 lists the options for the different hyper-parameters that we used.

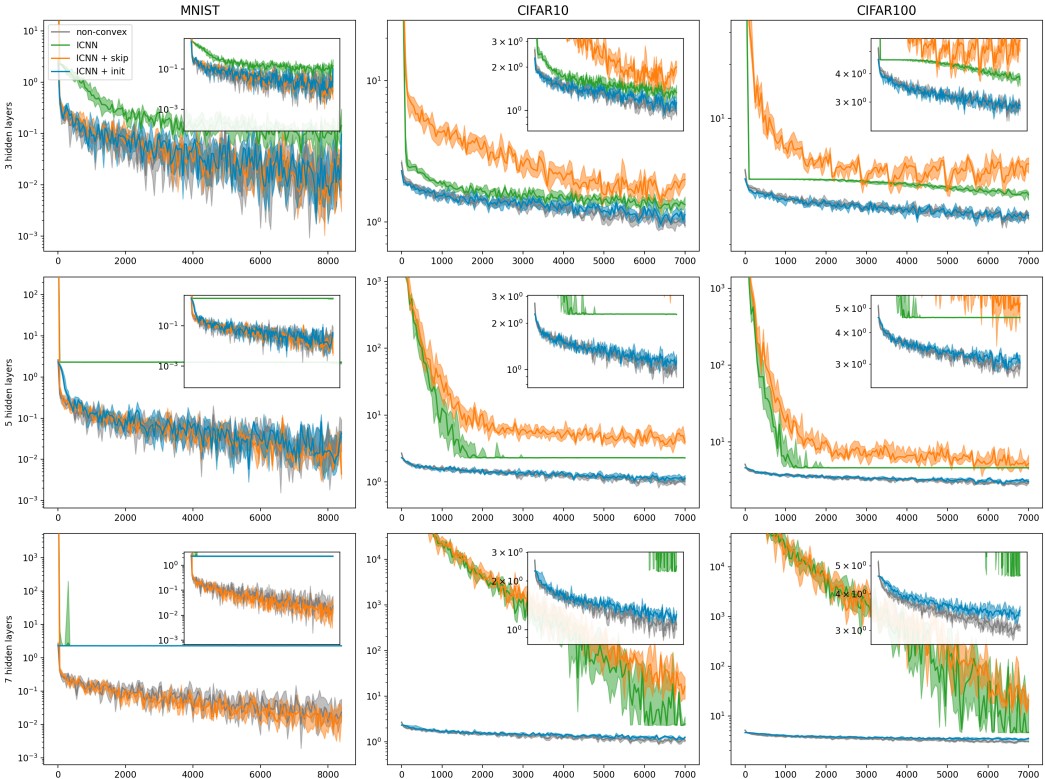

Figure 5: Training loss curves of ICNN variants with the same architecture with three, five and seven hidden layers on the MNIST, CIFAR10 and CIFAR100 datasets. "ICNN" is an Input-Convex Neural Network with He initialisation. "ICNN + skip": same settings but with skip-connections. "ICNN + init": our principled initialisation for ICNNs w/o skip-connections. "non-convex": a traditional non-convex network. The median performance over ten runs is displayed, shaded regions represent the inter-quartile range. The inset figures provide a view of the loss curves zoomed in.

Table 2: Hyper-parameter search space for results in Table 3

| pre-processing | | {none, PCA, ZCA} |
|---|---|---|
| learning rate | | $\{10^{-2}, 10^{-3}, 10^{-4}\}$ |
| $L_2$ regularisation | | $\{0, 10^{-2}\}$ |
| | 1 hidden | {(1k), (10k)} |
| layer-widths | 2 hidden | {(1k, 1k), (1k, 10k), (10k, 1k), (10k, 10k)} |
| | 3 hidden | {(1k, 1k, 1k), (1k, 10k, 1k), (10k, 1k, 10k), (10k, 10k, 10k)} |

Table 3: Optimal hyper-parameters for each configuration in Figure 3. The epoch column reports the epoch in which the validation accuracy was highest.

| | | architecture | $\eta$ | $L_2$ | pre-processing | epoch | accuracy |
|---|---|---|---|---|---|---|---|
| **MNIST** | non-convex | (784, 1 000, 1 000, 10) | 1e-4 | 0.01 | ZCA | 25 | 98.62% |
| | ICNN | (784, 10 000, 10) | 1e-4 | 0.01 | PCA | 24 | 98.28% |
| | ICNN + skip | (784, 10 000, 10) | 1e-3 | 0.00 | none | 24 | 98.30% |
| | ICNN + init | (784, 10 000, 10) | 1e-4 | 0.00 | ZCA | 15 | 98.27% |
| **CIFAR10** | non-convex | (3072, 10 000, 1 000, 10) | 1e-4 | 0.01 | ZCA | 18 | 55.92% |
| | ICNN | (3072, 1 000, 10) | 1e-4 | 0.00 | none | 19 | 54.74% |
| | ICNN + skip | (3072, 1 000, 10) | 1e-4 | 0.01 | none | 19 | 52.47% |
| | ICNN + init | (3072, 1 000, 10) | 1e-4 | 0.00 | none | 17 | 55.64% |

## C.3 Random Bias Initialisation

In section 3.3, we chose to set $\sigma_b^2 = 0$ for simplicity. However, it is also possible to derive an initialisation with $\sigma_b^2 > 0$. Practically, this means that we can initialise the biases with random samples from a Gaussian distribution with mean $\mu_b$ as in eq. (8) and $\sigma_b^2 > 0$. However, due to eq. (10) we know that this will also affect the initialisation of the synaptic weight parameters.

For the variance of the weight variance to be positive we must consider the following inequality (from eq. 37):

$$\sigma_w^2 = \frac{2}{1+\alpha^2} \frac{1}{N} \left( 1 - \frac{\sigma_b^2}{\sigma_*^2} - \mu_w^2 f_c \big( \mathrm{Corr}[s_1^-, s_2^-] \big) \right) > 0$$

$$\Leftrightarrow 1 - \mu_w^2 f_c \big( \mathrm{Corr}[s_1^-, s_2^-] \big) > \frac{\sigma_b^2}{\sigma_*^2}$$

$$\Leftrightarrow \sigma_*^2 \left( 1 - \mu_w^2 f_c \big( \mathrm{Corr}[s_1^-, s_2^-] \big) \right) > \sigma_b^2.$$

Assuming we are working in a fixed point regime, we obtain the following simplified constraint: $0 \le \sigma_b^2 < \sigma_*^2 (1 - \rho_*)$. Based on this constraint, we propose to specify the bias variance using $\sigma_b^2 = \beta(1 - \rho_*)\sigma_*^2, \beta \in [0, 1)$. As a result, we obtain the following expression for the variance of the weights:

$$\sigma_w^2 = \frac{2}{1+\alpha^2} \frac{1}{N} \left( 1 - \beta(1 - \rho^*) - \mu_w^2 f_c \big( \mathrm{Corr}[s_1^-, s_2^-] \big) \right).$$

Combining this result with equations (33) and (40) under similar conditions as in section B.3, we obtain the following initialisation parameters:

$$\mu_w = \pm\sqrt{\rho_* f_c(\rho_*)^{-1}} \qquad\qquad \sigma_w^2 = \frac{2}{1+\alpha^2} \frac{1}{N}(1 - \rho_*)(1 - \beta) \qquad (45)$$

$$\mu_b = -N\mu_w(1-\alpha)\sqrt{\frac{\sigma_*^2}{2\pi}} \qquad\qquad \sigma_b^2 = \beta(1 - \rho_*)\sigma_*^2. \qquad (46)$$

Note that this introduces an additional hyper-parameter, $\beta \in [0, 1)$.

To get an idea of the effects of setting $\beta > 0$, we ran an additional experiment with $\beta = \frac{1}{2}$. By introducing variation in the initial bias parameters, the pre-activations at initialisation can become negative as well. As a result, we suspect that the distribution of the pre-activations can become more Gaussian-like by increasing $\beta$. Figure 6 provides a comparison of the propagation with $\beta = 0$ and $\beta = \frac{1}{2}$, showing a subtle, yet visible, effect. Figure 7 shows learning curves for $\beta \in \{0, \frac{1}{2}\}$. Here, we observe that the random biases enables training of ICNNs in the 7-layer network on MNIST. This indicates that initialising the bias parameters with Gaussian random samples can further improve the model performance.

## C.4 Ablation Studies

To better understand the effects of the various hyper-parameters of our proposed method, we ran a set of ablation experiments. One of the most important factors of our proposed method is the initialisation of the bias parameters. This raises the question whether it might suffice to initialise the bias parameters (cf. eq. 8) and ignore the weights entirely. Figure 8 compares networks trained with and without our full initialisation to those where only the bias parameters are initialised according to eq. (8). The weight parameters for the latter networks are initialised using strategies for regular networks. Our experiments show that only centring the pre-activations does not suffice to make these networks learn better. The networks converge faster, but they get stuck on the same plateau that ICNNs that do not use our principled initialisation get stuck on. For shallow networks, the bias shift even seems to prevent ICNNs from getting past this plateau, leading to inferior learning (first row of Figure 8).

The choice for $\rho_* = \frac{1}{2}$ in section 3 is rather arbitrary. The main motivation for this choice is that it leads to a closed-form expression for the ReLU kernel (see section A.3). However, other choices for $\rho_*$ are still possible. Since the fixed point for $\rho_*$ is not stable (see section B.4), the correlation still tends to drift towards one. As a result, choosing a lower value for $\rho_*$ might allow to train even deeper networks. We verified this experimentally and the learning curves for these experiments can be found in Figure 9.

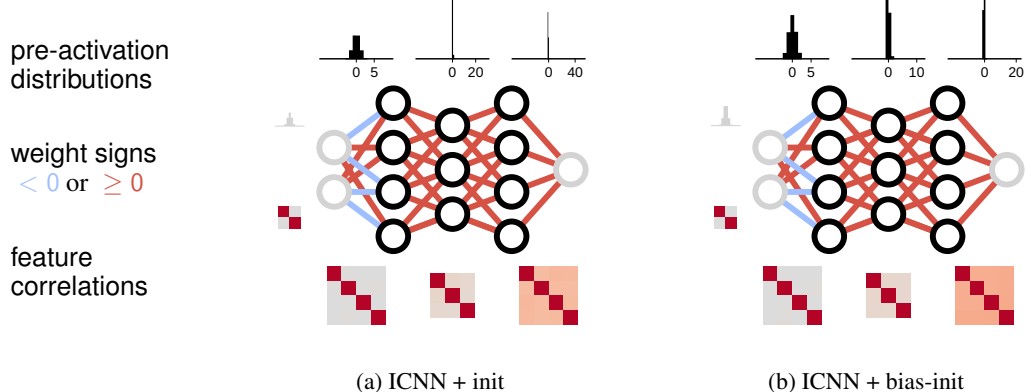

pre-activation distributions

weight signs
$< 0$ or $\geq 0$

feature correlations

(a) ICNN + init  (b) ICNN + bias-init

Figure 6: Illustration of the effects due to good signal propagation in hidden layers. Blue and red connections depict negative and positive weights, respectively. The distributions of pre-activations in each layer are shown on the top. On the bottom, the feature correlation matrices in each layer are displayed. The input distribution is depicted by the small elements on the left.

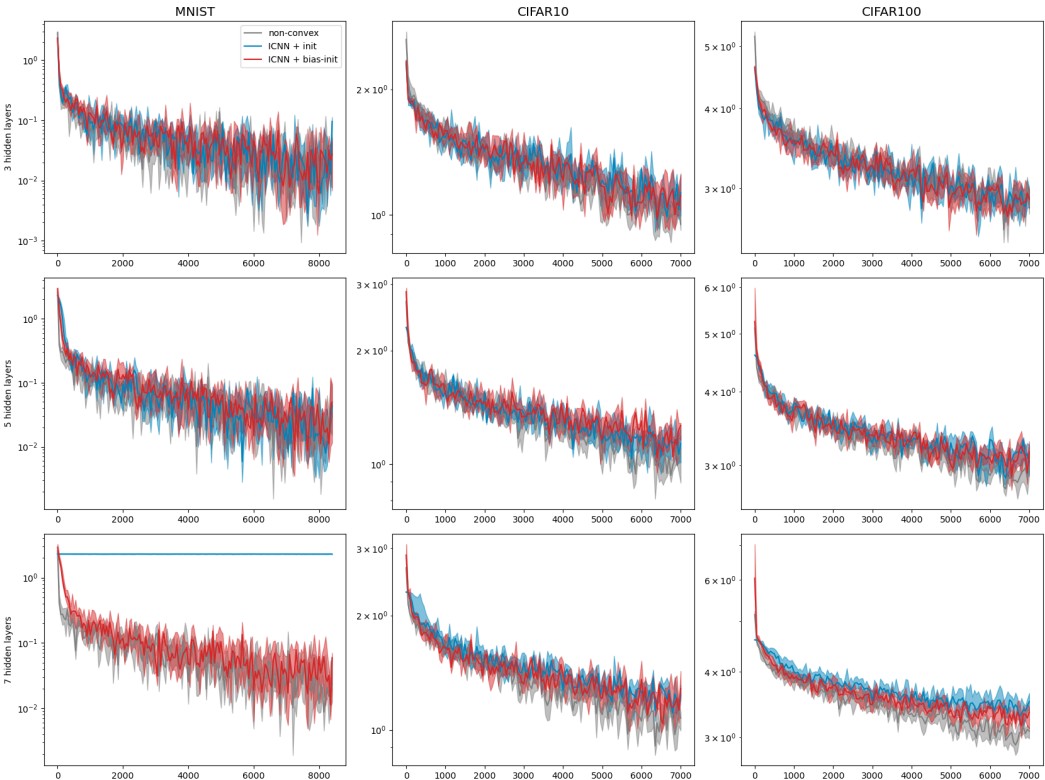

Figure 7: Training loss curves of ICNN variants with the same architecture with three, five and seven hidden layers on the MNIST, CIFAR10 and CIFAR100 datasets. "ICNN + init": our principled initialisation for ICNNs with $\sigma_b^2 = \beta = 0$. "ICNN + bias-init": our principled initialisation for ICNNs with $\sigma_b^2 = \beta = \frac{1}{2}$. "non-convex": a traditional non-convex network. The median performance over ten runs is displayed, shaded regions represent the inter-quartile range.

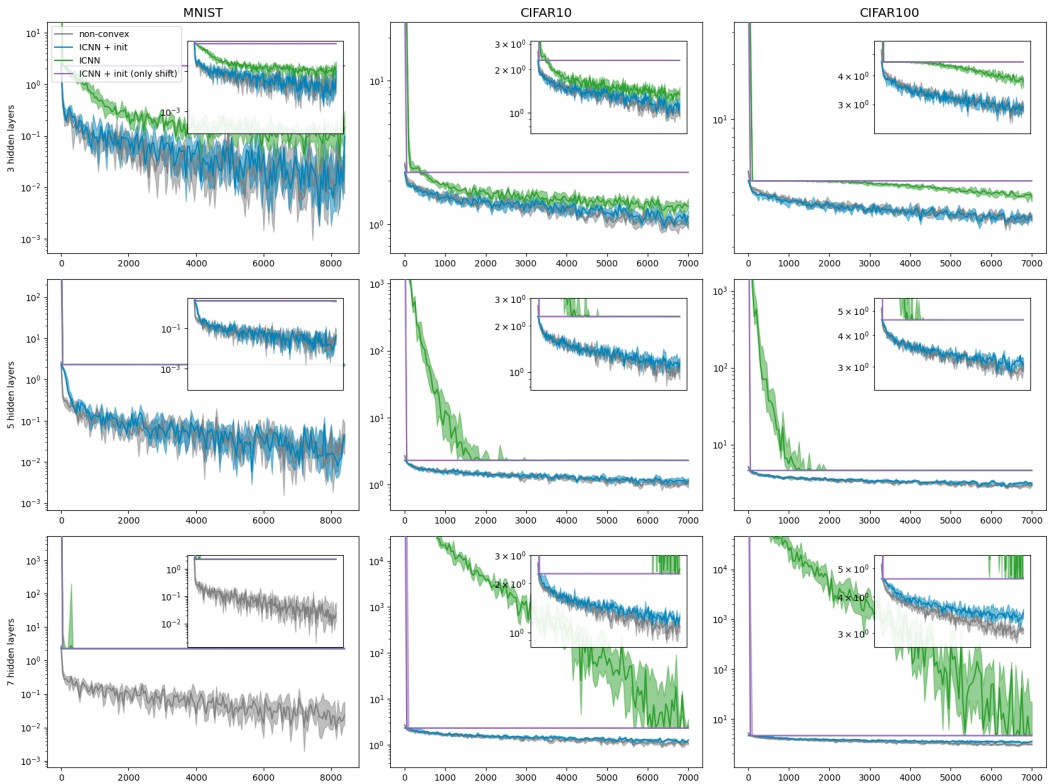

Figure 8: Training loss curves of ICNN variants with the same architecture with three, five and seven hidden layers on the MNIST, CIFAR10 and CIFAR100 datasets. "ICNN + init": our principled initialisation for ICNNs w/o skip-connections. "ICNN" is an Input-Convex Neural Network with He initialisation. "ICNN + init (only shift)": our principled initialisation for ICNNs for bias parameters only. "non-convex": a traditional non-convex network. The median performance over ten runs is displayed, shaded regions represent the inter-quartile range. Note that learning curves for "ICNN" and "ICNN + init (only shift)" overlap on MNIST.

## C.5   Different ICNN Implementations

The original ICNN uses a projection method after every update to keep the weights positive (Amos et al., 2017). An alternative method to keep weights positive is to re-parameterise the weights using a function with positive co-domain (cf. Nesterov et al., 2022). If weights are re-parameterised using the exponential function, we do not need to draw weights from a log-normal distribution. I.e. such that $\boldsymbol{W} = \exp(\tilde{\boldsymbol{W}})$, where $\exp(\cdot)$ is applied element-wise. Instead, the initial weights can be directly sampled from a Gaussian distribution. Figure 10 shows the result for the initialisation experiments from Figure 2 using the re-parameterisation instead of the projection method.

Figure 10 shows that our initialisation makes it possible to train ICNNs using the exponential re-parameterisation. This re-parameterisation greatly affects the learning dynamics of the network, however. Therefore, we find that this particular implementation of ICNNs generally does not perform quite as well as the original implementation. Figure 11 shows the learning curves for re-parameterised ICNNs using the hyper-parameters from table 4. The search space (in Table 2) is the same as for Figure 3 in the main paper.

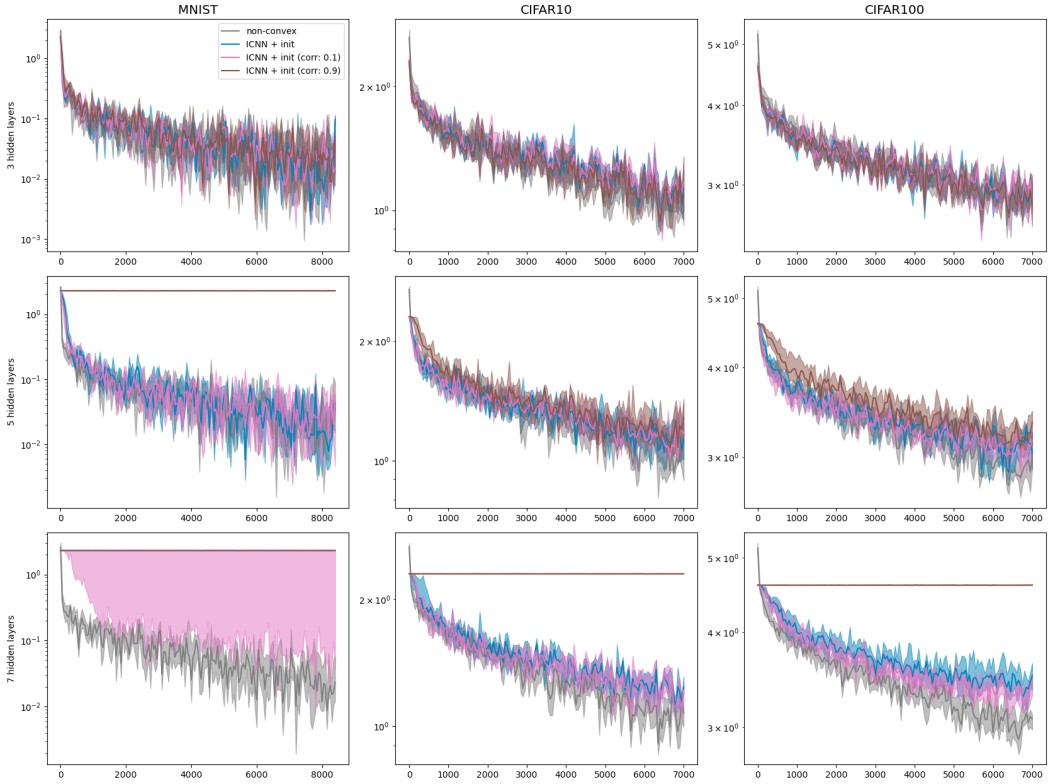

Figure 9: Training loss curves of ICNN variants with the same architecture with three, five and seven hidden layers on the MNIST, CIFAR10 and CIFAR100 datasets. "ICNN + init": our principled initialisation for ICNNs with $\rho_* = \frac{1}{2}$. "ICNN + init (corr: $r$)": our principled initialisation for ICNNs with $\rho_* = r$. "non-convex": a traditional non-convex network. The median performance over ten runs is displayed, shaded regions represent the inter-quartile range.

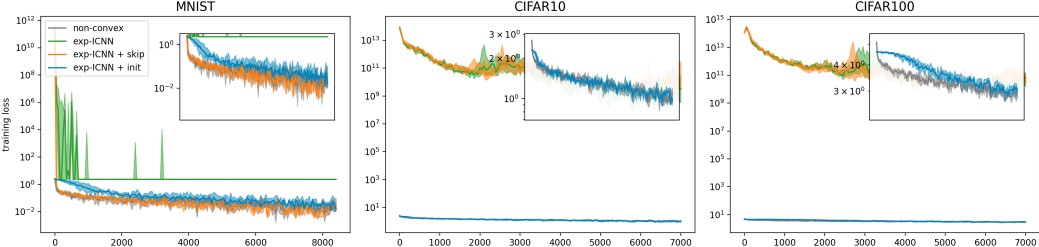

Figure 10: Training loss curves of ICNN variants using an exponential reparameterisation of weights. The reference exp-ICNN has no skip-connections and is initialised with the default He-initialisation. The exp-ICNN + skip model additionally includes skip-connections, but no principled initialisation. Our principled initialisation for exp-ICNNs without skip-connections is denoted by ICNN + init. The results for a non-convex network are included for reference. Each curve displays the median performance over ten runs. Shaded regions represent the region between the quartiles and dashed lines represent min- and maxima over the ten runs. The inset figures provide a view of the loss curves zoomed in on the reference non-convex network.

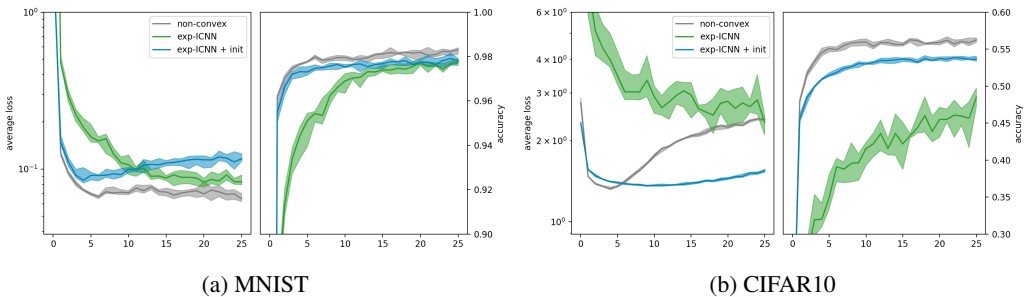

|             | (a) MNIST | (b) CIFAR10 |
|-------------|-----------|-------------|

Figure 11: Test set metrics of compared methods on the (a) MNIST and (b) CIFAR10 datasets. Each curve displays the median performance over ten runs. Shaded regions represent the inter-quartile range over the ten runs.

Table 4: Optimal hyper-parameters for each configuration in Figure 11. The epoch column reports the epoch in which the validation accuracy was highest.

|       |                | architecture             | $\eta$ | $L_2$ | pre-processing | epoch | accuracy |
|-------|----------------|--------------------------|--------|-------|----------------|-------|----------|
| MNIST | non-convex     | (784, 1 000, 1 000, 10)  | 1e-4   | 0.01  | ZCA            | 25    | 98.62%   |
|       | exp-ICNN       | (784, 1 000, 10)         | 1e-4   | 0.01  | none           | 25    | 98.02%   |
|       | exp-ICNN + init| (784, 1 000, 1 000, 10)  | 1e-2   | 0.00  | ZCA            | 25    | 98.18%   |
| CIFAR | non-convex     | (3072, 10 000, 1 000, 10)| 1e-4   | 0.01  | ZCA            | 18    | 55.92%   |
|       | exp-ICNN       | (3072, 1 000, 10)        | 1e-4   | 0.01  | none           | 22    | 48.38%   |
|       | exp-ICNN + init| (3072, 10 000, 10)       | 1e-3   | 0.00  | PCA            | 24    | 55.10%   |