# OpenReview forum: "Principled Weight Initialisation for Input-Convex Neural Networks"
_NeurIPS.cc/2023/Conference — NeurIPS 2023 poster_

### Official Review · Reviewer_Ryn9 · 2023-06-28

**Soundness:** 3 good
**Presentation:** 3 good
**Contribution:** 3 good
**Rating:** 6
**Confidence:** 4

**Summary:**

The paper discusses a principled weight initialization strategy for Input-Convex Neural Networks (ICNNs). The authors propose a new theory that generalizes signal propagation theory to include weights without zero mean, and derive a principled initialization strategy for ICNNs from this theory. They demonstrate the effectiveness of their initialization strategy through empirical experiments and apply ICNNs in a real-world drug-discovery setting.


**Strengths:**

1. The paper proposes a new theory that generalizes signal propagation theory to include weights without zero mean, which is a significant contribution to the field.
2. The authors derive a principled initialization strategy for ICNNs from their new theory, which improves learning and generalization in ICNNs.

**Weaknesses:**

1. The proposed approach cannot be applied to networks with skip connections, limiting its usage in real-world models.
2. The experimental section is not comprehensive and there is a lack of ablation study

**Questions:**

N/A

**Limitations:**

Yes

---

> ### Author Rebuttal · Authors · 2023-08-03
>
> We would like to thank you for your feedback. We hope to address all of your points in detail soon!
>
> For now, we would like to ask for some further clarification on what ablation studies are missing.
> This would help us to better address your concerns in our detailed response.
>
> Thanks in advance!
>
> ---
>
> We thank the reviewer for their positive and encouraging feedback.
> We hope that we can shed light on the mentioned weaknesses.
>
> ##  Skip connections
>
> Our proposed initialisation scheme was developed to enable faster training in ICNNs.
> Because skip-connections have a similar purpose, we would argue that our initialisation can be a replacement for skip-connections.
> Prior to our work, CNNs could not be trained without skip-connections.
> With our principled initialisation, we make it possible to train ICNNs without skip-connections.
> Furthermore, our initialisation appears to enable better results than ICNNs with skip-connections, which have more parameters.
>
> There is also a strong trend to replace skip-connections in regular networks.
> For example in (Zhang et al., 2022) it is shown that regular deep networks can be trained to the same performance as ResNets by carefully controlling signal propagation.
> Also, note that skip-connections in ICNNs usually connect layers with the input to the network and not block-wise, as in residual networks.
> Therefore, ICNNs used in prior work have a strong tendency towards dynamics that are similar to single-layer networks and suffer from the feature reuse problem (Zagoruyko et al., 2016).
> The ICNNs that we trained in this work have to develop feature hierarchies and are therefore a step forward for researchers working with ICNNs.
>
> Furthermore, we emphasise that our method can be applied to networks with skip-connections in practice.
> It is just the theory that is difficult to derive because of the dependency between the inputs and outputs of the residual branch.
>
> ### Additional References
>  - Zagoruyko, S., & Komodakis, N. (2016). Wide Residual Networks. Proceedings of the British Machine Vision Conference 2016, 87.1-87.12. https://doi.org/10.5244/C.30.87
>  - Zhang, G., Botev, A., & Martens, J. (2022). Deep Learning without Shortcuts: Shaping the Kernel with Tailored Rectifiers. International Conference on Learning Representations 10. https://openreview.net/forum?id=U0k7XNTiFEq
>
> ## Ablation study
>
> First of all, we would like to point out that our main experiments are already an ablation study.
> We start from ICNNs with skip-connections (the most common form of ICNN at the time of writing) and establish the baseline performance.
> The first ablation is obtained by removing the skip-connections which results in networks that become practically impossible to train.
> Finally, we add our principled initialisation to the networks without skip-connections and find that the initialisation improves not only the networks without skip-connections, but also the networks with skip-connections.
> Unfortunately, we were unable to deduct which ablations you would have liked to see in addition.
>
> We included learning curves for different choices of $\rho_*$ as suggested by reviewer f5Cq (figure 2c in the rebuttal PDF).
> We also investigated what happens if we choose to initialise the biases with $\sigma_b > 0$ (figure 2a in the rebuttal PDF).
> Finally, we added an experiment to see how important the effect of the bias shift is by only using eq. (8) for initialising the bias parameters.
> The weight parameters were initialised using an initialisation scheme for regular networks.
> The results of this experiment can be found in figure 2b of the rebuttal PDF.
> As suspected, only initialising the bias parameter already leads to good performance, but results tend to be better/more consistent when including the weights for the initialisation.

---

### Official Review · Reviewer_f5Cq · 2023-07-02

**Soundness:** 2 fair
**Presentation:** 2 fair
**Contribution:** 2 fair
**Rating:** 3
**Confidence:** 3

**Summary:**

The proposed method aims to solve the initialisation problem in an Input convex Neural Network (ICNN), where the weights are required to the non-negative. The commonly applied approach, setting the negative entries stamped from zero-mean Gaussian as zero, leads to varies the desired mean. By analysing the signal propagation in the Neural Networks, the authors are able to sample the weights with expected mean and variance. The method is evaluated on several tasks and compared with the ICNN without the proposed initialisation method with some promising results.

**Strengths:**

1. The problem solved in the submission is important in the ICNN setting.
2. The derivation of the method is concrete.

**Weaknesses:**

Some details are not well explained and are sometimes confusing.
1. $\rho_*$ is defined $\frac{1}{\sigma^2_*}Cov[s_1^-, s_2^-]$ in line 178 and the author claim $\rho_*$ is independed on $\frac{1}{\sigma^2}$ in line 184. And then  \rho_* is set as $\frac{1}{2}$ heuristically.

The experiments are not carefully designed to support the submission and I am not sure some experiments are sufficiently conducted.
1. In Figure 2, the loss of ICNN does not change during the training process in the MNIST setting but in CIFAR10 and CIFAR100 there is no such phenomenon.
2. In Figure 3, drawing the conclusion that with the proposed initialisation the training dynamic is more stable is not concrete via eyeball comparison. Some quantities are needed.
3. And according to my experience of training non-convex NN on cifar10, the average testing loss does not really show a dramatic increase after some training iterations and non-convex NN has the highest test loss and accuracy which are unusual. And the rest of the learning curve is more reasonable with low test loss and corresponding high accuracy.
4. It can be noticed that with skip on ICNN the performance of ICNN improves, I think applying the proposed initialisation to ICNN with skip will give good support to the submission.

Additional question, which does not affect my score. Since in all the settings in the submission, the non-convex NN has the best performance compared with other ICNN-based methods, so is there any scenario for ICNN or why ICNN is essential in the first place?

Notations:
In some equations the $s_i$, $s_j$ are mixed used with $s_1$, $s_2$, for example, the one below line 172 and the one below 175.

**Questions:**

1. In Figure 2, Non-convex and ICNN have lower training loss than other baselines at iteration 0 which, I believe, indicates the initialisation can the author explain why it happens?
2. Can we treat $\rho_*$ as a hyperparameter? If so how it affects training and generalisation of the ICNN models? these two questions are not discussed.

**Limitations:**

See the sections above.

---

> ### Author Rebuttal · Authors · 2023-08-03
>
> We would like to thank you for your feedback. We hope to address all of your points in detail soon!
>
> For now, we would like to ask for some further clarification on what quantities the reviewer would like to see in the comparison.
> This would help us to better address your concerns in our detailed response.
>
> Thanks in advance!
>
> ---
>
> We thank the reviewer for their critical feedback and apologise for details that were confusing or unclear.
> We try to clarify these details better in the answers below and the final version.
> We hope that you will take the time to consider our rebuttal and are willing to update your score after possible further discussions.
>
> ## Weaknesses
>
> Concerning the unclarity:
>
> 1. We agree that the statement might be confusing, but it is inherent to the definition of correlation.
> Because the correlation is defined in terms of the variance they can not be independent.
> Our statement about eq. (11) and (12) being independent aims to make clear that once we have computed the correlation, we do not explicitly need the variance to derive our initialisation.
> We reformulated line 184 to avoid confusion.
>
>    The choice for setting $\rho_* = \frac{1}{2}$ can indeed be considered as a heuristic.
>    Our main motivation was to get a closed-form result for $\arccos(\rho_*)$, but other values are possible as well (more details below).
>    We will add a section in the appendix with these details in the final revision.
>
> Concerning the experiments:
>
> 1. Note that the loss for ICNN does actually go down during the first hundred update steps, but we agree that this is barely visible in the plot.
> The CIFAR-10 and CIFAR-100 plots for ICNNs without skip-connections or initialisation show the same behaviour: the training loss goes down initially, but stops improving at some point.
> We suspect that this happens because the network attempts to reduce the activation strength of the network by pushing weights and biases down.
> In the example of the MNIST models, the biases in the first unconstrained layer soon become all negative.
> As a result, the ReLU activations, which are the inputs to the first constrained layer, are all zero and the network can not learn a function from the inputs.
> The skip-connections alleviate these issues by bypassing these dead layers.
> Our initialisation simply provides a better starting point where activations do not need to be reduced to improve the error early in training.
> We will try to include a more elaborate discussion in the final version of the paper.
> 2. We could not find the point where we conclude that training dynamics are more stable in the context of Figure 3.
> Figure 3 aims to show that our initialisation does not only affect the empirical error but also translates to generalisation performance.
> We agree that there is little to no difference for the MNIST experiments and we also acknowledge that in our manuscript (line 283).
> However, we would argue that for the CIFAR10 experiments, our method (ICNN + init) clearly leads to faster learning compared to other ICNNs.
> The quantitative accuracies for the validation runs can be found in Table 3 in the appendix.
> 3. The increase in the test loss in Figure 3 can be explained by overfitting on the training and/or validation data.
> Note that the optimal models were chosen based on validation accuracy, as indicated by Table 3 in the appendix.
> Because the loss is only a proxy for accuracy, the optimal model might indeed be in an overfitting regime in terms of test loss.
>
>    By revisiting Table 3, we realised that Figure 3 does not depict the early stopping.
>    We updated the figure for the final version (Figure 3 in the rebuttal PDF).
> 4. One of our contributions is to show that ICNNs do not need skip-connections if they are initialised in a principled manner.
> This indicates that skip-connections mainly help in making ICNNs trainable.
> We did (accidentally) run the Tox21 experiments with both skip-connections and our initialisation and did not observe any substantial improvements over using only our initialisation.
> However, the skip-connections do effectively modify the signal propagation in a non-trivial way.
> It would require additional analysis to obtain a principled approach for initialising networks with skip-connections.
> We also refer to a similar reply we gave to reviewer Ryn9 with more references in this context.
>
> Concerning your additional questions:
>
>  - The performance of ICNNs should not be compared directly with regular networks.
> As indicated by reviewer kLTe, ICNNs have theoretically less capacity.
> ICNNs have the unique property that they are convex.
> This can be useful or is even necessary in various settings (e.g. energy-based models, optimal transport, level-set exploration, …) that we describe in our related work section.
>  - The notation on lines 172 and 175 is on purpose.
> We aim to compute the (co)variance for arbitrary $s_i$ and/or $s_j$ on the left-hand side.
> Under our assumptions, the (co)variance turns out to be independent of the index $i$ or $j$.
> We explicitly use $s_1$ and $s_2$ on the right-hand side to emphasise this independence.
>
> ## Questions
>
> 1. The ICNNs have only non-negative weights.
> Due to the ReLU activation function, also activations will be positive.
> Computing these dot products typically leads to numerically large values.
> This is also why the loss will typically be large.
> Our initialisation counters these effects mainly by initialising the bias parameters with negative values.
> 2. $\rho_*$ can indeed be treated as a hyper-parameter.
> Because of the tendency of the correlation to grow as the network grows deeper, lower $\rho_*$ values will typically make it possible to train deeper networks.
> We ran ablation experiments on the choice for $\rho_*$ to obtain figure 2b in the rebuttal PDF.
> As expected, lower values for $\rho_*$ can enable deeper networks, but overall, performance is very similar.
> We will include these results in the appendix of the final version.

---

### Official Review · Reviewer_YxjJ · 2023-07-06

**Soundness:** 3 good
**Presentation:** 4 excellent
**Contribution:** 3 good
**Rating:** 6
**Confidence:** 4

**Summary:**

This paper investigates the initialization for input-convex neural networks. They generalize the signal progagation theory by removing the assumption of centred weight distrubution. The experiments show that the proposed initiaization method is effective in a set of datasets.

**Strengths:**

1 This paper generalizes signal propagation theory by removing the assumption that weights are sampled from a centred distribution. This generalization is necessary for ICNNs.

2 The experimetal results are solid and convincing. The experiment on the real-world drug discovery task is nice.

**Weaknesses:**

1 The theoretical contribution of this paper is limited. The theoretical results are derived  following the framework proposed in [1].

2 It seems that the authors only consider the forward propagation of an initial ICNN. However, the backward propagation and the output diversity are also crucial for the initialization.






[1] Samuel S Schoenholz, Justin Gilmer, Surya Ganguli, and Jascha Sohl-Dickstein. Deep information propagation. In International Conference on Learning Representations, 2017

**Questions:**

Please see weaknesses.

**Limitations:**

The authors havve adequately addressed the limitations, and there is no potential negative societal impact of their work.

---

> ### Author Rebuttal · Authors · 2023-08-09
>
> ## Theoretical contribution
>
> We do not entirely understand why our generalisation of signal propagation theory and the derivation of the initialisation for ICNNs would be considered a _”limited”_ theoretical contribution.
> The initialisation for ICNNs is indeed _limited_ to ICNNs.
> However, the generalised signal propagation theory could be useful outside of the scope of ICNNs as well.
> For example, it could be used to study certain effects in regular networks due to the initial weights of a network not having exactly zero mean in practice.
> After all, our theory shows that feature correlation affects signal propagation as soon as the weights do not have zero mean.
>
> As mentioned in the introduction and section 2 of our manuscript, we build on the framework presented in the thesis of Neal (1995), which was also the basis for the work from Poole et al. (2016).
> The main contribution to signal propagation from Poole et al. (2016) was to include the correlation between samples in the analysis.
> The work from Schoenholz et al. (2017) extends the propagation in (Poole et al., 2016) by including depth scales, dropout and the backward pass.
> Further work extended the propagation for resnets (Yang et al., 2017) and convolutional layers (Xiao et al., 2018).
> This information can also be found in the related work section of our manuscript.
> In this context, we believe that our extension of signal propagation theory is not more _limited_ than any of these published works.
> Furthermore, we believe that our derivation of an initialisation for ICNNs is also not more _limited_ than e.g. the one that was recently derived for hypernetworks (Chang et al., 2020).
>
> Finally, our work does not include the correlation between samples, in contrast to (Poole et al., 2016; Schoenholz et al., 2017).
> We show that the variance in a layer not only depends on the variance but also on the correlation between features in the previous layer when weights have non-zero mean.
> As a result, the propagation of correlation between features interferes with the propagation of variance, which is arguably more complex than studying independent signals as in (Poole et al., 2016; Schoenholz et al., 2017).
> Note that although the expressions are very similar, the correlation between samples and the correlation between features are two different things.
> In this sense, we would argue that our work is not _limited_ to a simple derivation of (Poole et al., 2016) or (Schoenholz et al., 2017), but rather provides a different perspective into signal propagation theory.
>
> ## Backward Analysis
>
> We agree that incorporating the analysis of the backward pass to derive an initialisation is generally desirable.
> Therefore, we decided to include the generalised signal propagation of the backward pass in the appendix.
> The mean and variance propagation of the deltas in backpropagation (using similar assumptions as Schoenholz et al. (2017) is given by
> $$\begin{align*}
> 	\mathbb{E}[\delta_j^{-}] &= M \mu_w \mathbb{E}[\phi'(s_1^{-})] \mathbb{E}[\delta_1] \\\\
> 	\mathbb{E}[\delta_i^{-} \delta_j^{-}] &= \updelta_{ij} M \sigma_w^2 \mathbb{E}\bigl[\phi'(s_1^{-})^2\bigr] \mathbb{E}\bigl[\delta_1^2\bigr] + \mu_w^2 \mathbb{E}[\phi'(s_i^{-}) \phi'(s_j^{-})] \sum_{k,k'} \mathbb{E}[\delta_k \delta_{k'}],
> \end{align*}$$
> Where $\updelta_{ij}$ is the Kronecker delta and $\delta_i = \frac{\partial L}{\partial s_i}$.
> Note that, similar to the forward pass, the covariance between entries in the delta vectors interferes with the variance propagation.
> The kernel for the derivative of $\operatorname{LReLU}$ with parameter $\alpha$ is given by
> $$\mathbb{E}[\operatorname{LReLU}'(s_1 \mathbin{;} \alpha) \operatorname{LReLU}'(s_2 \mathbin{;} \alpha)] = (1 - \alpha)^2 \frac{1}{2 \pi} \arccos(-\rho) + \alpha.$$
> This should make it possible to derive initialisations that also incorporate the backward pass, e.g. using appraoches from (Glorot et al., 2010) or (Defazio et al., 2021).
>
> ## Experiments
>
> We thank you for the positive assessment of our experimental section.
> We also think that the latent-space exploration of molecules is a particularly illustrative example of the usability of ICNNs.

---

### Official Review · Reviewer_kLTe · 2023-07-24

**Soundness:** 3 good
**Presentation:** 3 good
**Contribution:** 3 good
**Rating:** 7
**Confidence:** 3

**Summary:**

NOTE: edited after author rebuttal and score has been updated.

This paper is related to input convex neural networks (ICNN). It analyzes the signal propagation through such a network and based on that proposes a new initialization scheme that allows the networks to be trained efficiently. It investigates the efficacy of the scheme in various ML tasks, including image benchmarks and drug discovery tasks.


**Strengths:**

The initialization scheme is well grounded in theory (although with some stated simplified gaussian assumption).

The method seems to work well in practice, at least in the tasks used in the paper.

It is experimentally shown that in contrary to previous work claims, ICNNs do not require skip connections in order to be trainable to good results.


**Weaknesses:**

The ICNN model family is intuitively less powerful than normal neural networks. The authors do not discuss this and the chosen experiments seem to be such that the performance degradation compared to normal neural networks does not happen. Could the authors discuss the limitations of the ICNN models, e.g., could imagenet level or GPT-style text understanding models be trained in the ICNN setting?

The abstract says that the new initialization allows for more efficient latent space exploration, but as far as I see, there is no comparison to other methods in the paper.

Figure 1 shows preactivation distributions on the top, but the scale of the x axis is missing, please add that to the figure. It also seems that the activations start to concentrate around 0 as one moves deeper in the network. Is this the case and could the authors discuss this phenomenon and how this will affect very deep networks (e.g., 20 layers).

In the experimental sections, it seems that latent space exploration is the main use case enabled by the ICNN and the new init method? Could the authors discuss other use cases of ICNN and for example explain why they did not repeat the experiments from previous ICNN papers and show improvements stemming from their improved initialization? Also this use case seems very interesting, could the authors discuss pros and cons related to other methods that could be used to explore the latent space of this task? Also, are there some limitations of ICNN in this task or similar tasks, e.g., related to complexity or size of the network and the data?


**Questions:**

Minor spelling mistake 3.2 ”sufficiently good to the derive” - delete ”the”?


**Limitations:**

As mentioned in the paper, the signal propagation model assumes gaussian distributed preactivations, which is not always what happens in reality. This is well noted in the paper, and at least in the practical tasks considered in the empirical section, the derived initialization scheme still works well.

Although the authors say that it would be possible to analyze skip connections, they do not perform the analysis. The reasoning that skip connections are not needed might not hold for deeper networks.

---

> ### Author Rebuttal · Authors · 2023-08-09
>
> ## Limitations and Strengths of ICNNs
>
> You are right that the ICNN family is strictly speaking less powerful than regular networks because they are constrained to have convex decision boundaries.
> Note that it is possible to construct a universal approximation theorem using theorem 1 from (Yuille and Rangarajan, 2001), which has been used in (Sankaranarayanan and Rengaswamy, 2022).
> Our manuscript might indeed give the (misleading) impression that ICNNs are direct competitors of regular networks.
> However, ICNNs are supposed to be used if the convexity provides additional benefits or is required for a method to work.
> We provide the comparison with regular networks to emphasise that ICNNs are not necessarily harder to train than regular networks when using a principled (e.g. our proposed) initialisation.
> As the reviewer points out, there is indeed no ICNN that reaches the quality of state-of-the-art regular networks at ImageNet or the text understanding quality of GPT.
> However, we do believe that the difficulty of training ICNNs is slowing research in this direction down and that our initialisation method might make ICNNs a more viable alternative.
>
> We have added a few sentences to the experiment section to emphasise the goal of our comparisons with non-convex networks and to reduce this possible source of confusion.
>
> ## Choice of Experiments
>
> Indeed, we show the latent space exploration for molecular generation as a particular example, in which ICNNs could be relevant.
> Notably, we do perform experiments from other ICNN papers, concretely the computer vision benchmarks were also included in (Sivaprasad et al., 2021).
> For further examples where ICNNs are useful, we refer to our related work section.
> It might also be important to point out that our initialisation merely allows for more efficient training of ICNNs and is not the enabling component for these experiments.
>
> Our motivation behind these experiments  are some initial results by Nesterov et  al. (2022) who use ICNNs to “encourage” the latent space of an auto-encoder to build convex decision boundaries which allow efficient exploration of level sets.
> Several more methods for latent space exploration are compared in Du et al. (2023).
> They introduce a new method, ChemSpaceE, for latent space exploration, which does not use ICNNs, and a way to evaluate these methods.
> However , we did not manage to reproduce their results and were therefore unable to include meaningful comparisons.
> Furthermore, these other methods for latent space exploration do not allow traversing level-sets, which is eventually the main feature provided by ICNNs and the property that we chose to highlight in our final experiment.
>
> We will try to rephrase section 5.3 to resolve these sources of confusion:
>  - the motivation for this particular experiment
>  - the fact that ICNNs enable the level-set exploration, not our initialisation.
>
> ### Additional References
>
> Du, Y., Liu, X., Shah, N. M., Liu, S., Zhang, J., & Zhou, B. (2022). ChemSpacE: Interpretable and Interactive Chemical Space Exploration. Transactions on Machine Learning Research.
>
> ## Other Comments
>
> The scale of the x-axis in figure 1 is implicit in the width of the bins, which is constant for all histograms in a sub-plot.
> However, we agree that this should have been explained and an explicit scale indication is the better way to communicate this information.
> The activations tend to zero because any pre-activation that becomes negative is mapped to zero when using ReLU non-linearities.
> We found that this can be alleviated by initialising the bias parameters with non-zero variance.
> Due to eq. (10), this also requires a different weight variance to keep the same propagation.
> On the other hand, the instability of the fixed point (see appendix A.4) can lead to drift effects in the correlation, which effectively makes it hard to stabilise the propagation in very deep networks.
> Finding a setting where the correlation fixed point is actually stable would resolve this issue, but remains a problem for future work.
>
> We have updated figure 1 to provide an indication of the scale of the x-axis for the final version (figure 1 in the rebuttal PDF).
> We will also include a discussion about initialising the bias parameters with random samples drawn from a Gaussian distribution with variance $\frac{1}{2}$.
> We repeated the experiments from figure 5 in the appendix for this setting and find that the results are practically the same.
> Figure 2a in the rebuttal PDF shows the comparison between the initialisation with constant and random bias initialisation.
> These results also suggest that the randomness in the biases might enable training even deeper networks.

---

> > ### Comment · Reviewer_kLTe · 2023-08-15
> > **Thank you**
> >
> > Thank you for the rebuttal. My main concerns have been addressed and this has been reflected in the edited score.

---

### Author Rebuttal · Authors · 2023-08-09

We thank the reviewers for their feedback and suggestions.
The comments and concerns of each reviewer will be addressed individually and the paper will be updated correspondingly.

We did observe a common (potential) misunderstanding about our contribution in the reviews.
Some reviewers claim in their summary that we _“analyse”_ the signal propagation to derive an initialisation.
Although _analyse_ captures the detailed study of signal propagation, we believe it does not quite capture the value that we believe to have added to the traditional theory.
Therefore, we would like to emphasise that we do not just use signal propagation theory as a tool to derive an initialisation.
Instead, we generalise the existing signal propagation theory to allow initial weights with non-zero means.
Only then do we have the signal propagation tools to derive an initialisation for ICNNs.
Possibly this was already clear for the reviewers, but we want to avoid any misunderstandings.
A detailed discussion can be found in the reply to reviewer YxjJ.

We look forward to further feedback during the discussion phase.

---

### Comment · Area_Chair_KL8k · 2023-08-18
**Rebuttal acknowledgment**

Dear authors,

Thank you for your efforts in writing a rebuttal. Unfortunately not all reviewers have acknowledged or responded to it, but rest assured that I have read it and will bring it up in further discussion with the reviewers, and will take it into account for the final recommendation.

Best,

AC

---

### Decision · Program_Chairs · 2023-09-21

**Decision:**

Accept (poster)

**Comment:**

While its rating puts it squarely in the borderline zone, this paper makes a sound and sufficiently novel (if somewhat narrow) contribution. It appears that some of the reviewers with lower ratings might have misunderstood core aspects of the paper, which the authors addressed in a strong and thoughtful rebuttal that unfortunately went unanswered. Although additional experiments would have been useful, I side with reviewer kLTe in accepting the existing results as sufficient to validate the main claim of the paper: that the proposed framework provides a principled, efficient, and useful approach to initialize ICNNs.